# On the Patriarchal Lineages of Vinaya Transmission Starting with Upāli: Narratives and Interpretations in the Vinaya School 律宗 in China and Japan

**Weilin Wu**

School of Foreign Languages, Sun Yat-sen University, Guangzhou 510275, China; wuwlin@mail.sysu.edu.cn

**Abstract:** In both Pāli and Chinese vinaya literature, there are various patriarchal lineages of vinaya transmission in which Upāli is honored as the first patriarch. These lineages that start with Upāli can be categorized into two types. The first type is found mainly in Indian vinaya texts, including two groups of texts: the *Mohe sengqi lü* 摩訶僧祇律 (Skt. *Mahāsāṃghika-vinaya*), and the *Samantapāsādikā*, a Pāli *Vinaya* commentary, as well as its parallel Chinese version, the *Shanjianlü piposha* 善見律毗婆沙. The second type was constructed by Chinese Vinaya school masters in the Northern Song dynasty, who aimed to establish an orthodox Indian origin for the Vinaya school. After their introduction into China and Japan, the first type of lineages experienced transformation in later Vinaya school works composed by medieval Chinese and Japanese Buddhist monks. A comparative philological study on the *Samantapāsādikā* and *Shanjianlü piposha* shows a "mistranslated" Tanwude 曇無德 (Skt. Dharmagupta) in the patriarchal lineage of vinaya transmission in the *Shanjianlü piposha*, the parallel of which is "Buddharakkhita" in the Pāli sources. Further investigation on the Vinaya school reveals that both Dingbin 定賓 and Gyōnen 凝然, monks from the Vinaya school in later periods, identified the *Shanjianlü piposha* as a commentary on the *Dharmaguptaka-vinaya*, and they consequently considered the patriarchal lineage in the *Shanjianlü piposha* as the patriarchal genealogy of the Dharmaguptaka school, with the purpose of establishing an orthodoxy of the Vinaya school that could be traced back to Upāli. Furthermore, in the genealogy in the *Mohe sengqi lü*, Gyōnen associated the master Fahu 法護 with the Dharmaguptaka school. Yuanzhao 元照, an eminent Vinaya school monk, criticized the second type of lineages as false construction. Instead, he established a patriarchal lineage that starts with Tanwude, the editor and compiler of the *Dharmaguptaka-vinaya*, for the Chinese Vinaya school.

**Keywords:** Upāli; the Dharmaguptaka school; patriarchal lineages of vinaya transmission; Pāli sources; the Vinaya school



## 1. Introduction

Patriarchal worship plays a significant role in Chinese Buddhism, which originated in Indian Buddhism and was further developed in China. Chan Buddhism is one of the Chinese Buddhist schools in which patriarchal tradition is honored most, and that later influenced other Chinese Buddhist schools. In Chan Buddhism, there exists a will to orthodoxy in the construction of patriarchal lineage beginning with an Indian master, Bodhidharma 菩提達摩. Regarding patriarchal worship in Chan Buddhism, scholars such as Bernard Faure and John R. McRae have conducted a great number of studies.[1] Will to orthodoxy is also shown in the construction of patriarchal lineages in China's vinaya tradition, particularly the later dominant Vinaya school based on the *Dharmaguptaka-vinaya*? Ann Heirman has published an article on the early history of the Dharmaguptaka school, in which she traces it from its beginnings to the Tang dynasty in China. She provides a clear and useful survey of some important sources relevant to the history of the Chinese Dharmaguptaka tradition (Heirman 2002). Jinhua Chen conducted a detailed survey on the earlier Chinese vinaya patriarch Zhishou 智首 and his contemporaneous vinaya

specialists in connection with these predecessors, and reappraised Zhishou's historical position against the backdrop of a reconstructed history of the early vinaya tradition in China. His scholarship has clarified confusion surrounding some of the early Chinese vinaya patriarchs and their interrelationships (Chen 2017). In another article, Jinhua Chen performed an informative investigation on the lineage of the Chinese Vinaya school beginning with Facong 法聰 presented by Yan Zhenqing 顏真卿, a Tang bureaucrat, general, and calligrapher, in his *Fuzhou Baoying si Lüzang yuan jietan ji* 撫州寶應寺律藏院戒壇記 (*A Record of the Precept-platform in the Cloister of the Precept Treasure at the Baoying Temple in Fuzhou*) (Chen 2020). These studies have shed light on the earlier development of the Dharmaguptaka school in the Sui–Tang vinaya history.

However, the aforementioned studies do not focus on the Chinese vinaya masters' construction of the Indian patriarchal tradition in their engaging but historically unreliable myths in their sectarian narratives. This article will investigate these sectarian narratives and reveal the Chinese vinaya masters' will to orthodoxy in their construction of patriarchal tradition. My opinion is that one of the most typically Chinese features of the Vinaya school, which claimed to derive from the Indian patriarch Upāli, or Dharmagupta, was its insistence on a patriarchal tradition.

Upāli, one of the ten chief disciples of the Buddha, according to early Buddhist texts, is the person in charge of reciting and reviewing monastic discipline at the First Buddhist Council. However, according to vinaya literature, Upāli is also the first person in the lineage to transmit the Buddhist vinaya. There are two types of patriarchal lineages of vinaya transmission, starting with Upāli. The first type is descended from an Indian origin, as recorded in the Pāli and Chinese vinaya literature, including the *Samantapāsādikā*, a Pāli *Vinaya* commentary, and its parallel Chinese version, the *Shanjianlü piposha* 善見律毗婆沙, as well as in the *Mohe sengqi lü* 摩訶僧祇律 ascribed to the Mahāsāṃghika school. The second was constructed by monks from the Chinese Vinaya school 律宗. However, there is little research that discusses these types of patriarchal lineages starting with Upāli, or that probes their origin and transformation in the context of Chinese and Japanese Buddhism. Scholars in Indian Sanskrit and Pāli Buddhist studies have not paid attention to this issue in the context of East Asian Buddhism. Conversely, scholars in Chinese Buddhist studies have hardly used the relevant Pāli sources to investigate the origin and development of the patriarchal lineage, starting with Upāli, in the *Shanjianlü piposha*, nor have they paid attention to the *Mohe sengqi lü* or the second type of patriarchal lineages of vinaya transmission, starting with Upāli.

This paper examines how monks from the Vinaya school in China and Japan interpreted the Indian origin of the Dharmaguptaka school 法藏部 and made the *Sifen lü* 四分律 an authority based on the first type of lineages. By investigating the second type of lineages and relevant criticism from Yuanzhao 元照 in the Northern Song dynasty, this research also examines the Nanshan Vinaya school's 南山律宗 interpretation of the historical development of Buddhist vinaya. Throughout this study, we can see the Vinaya school masters' sectarian views on Indian Buddhism, and we thereby gain a deeper understanding of the development of the Vinaya school in China and Japan.

## 2. The Patriarchal Lineages of Vinaya Transmission Starting with Upāli in Pāli and Chinese Vinaya Literature

### 2.1. The Patriarchal Lineages of Vinaya Transmission in the Shanjianlü piposha and Samantapāsādikā

The *Shanjianlü piposha* (*Taishō* 1462) is one of the most important vinaya commentaries in China. Between 488 and 489 A.D., it was translated into Chinese in Guangzhou by a foreign monk named *Sengjiabatuoluo* 僧伽跋陀羅 (Saṅghabhadra, dates of birth and death unknown), and co-translated by Sengyi 僧猗 (dates of birth and death unknown). The *Samantapāsādikā* is a commentary on the Pāli *Vinaya*, written by the end of the fourth century or the beginning of the fifth century and traditionally ascribed to the commentator Buddhaghosa. In 1896, J. Takakusu initially stated that the *Shanjianlü piposha* is a translation of the *Samantapāsādikā*, and found that the Chinese translation corresponds, in general, to

the Pāli text of Buddhaghosa (Takakusu 1896). After Takakusu, two other Japanese scholars, M. Nagai and K. Mizuno, made further efforts to compare both texts. M. Nagai assumed that the original Indic text of the *Shanjianlü piposha* could not be the *Samantapāsādikā* we see today, for there are many terms transliterated from Sanskrit rather than Pāli in the Chinese version (Nagai 1922, pp. 69–133). K. Mizuno considered that the *Shanjianlü piposha* might be a translation of the Pāli *Samantapāsādikā*, though the former is much shorter than the latter (Mizuno 1937, 1938). P. Demiéville pointed out that the *Shanjianlü piposha* might be a translation of a prototype of the *Samantapāsādikā*, rather than the translation of the Pāli text as we know it today (Demiéville 1951). F. Lottermoser also proposed that the *Shanjianlü piposha* is a translation made from a version of the vinaya commentary that is different from the *Samantapāsādikā* as we see it now (Lottermoser 1982, p. 163). H. Bechert supports Lottermoser's proposal. He remarks that the differences between both texts indicate that it seems impossible that the extant Pāli *Samantapāsādikā* was the direct source of the *Shanjianlü piposha*, despite their relatively close correspondence (Bechert 1986, p. 138). By studying the title "Shanjianlü piposha" and terms transliterated from Sanskrit, as well as the structure of the Chinese version, Ananda W. P. Guruge proposed that the origin of the *Shanjianlü piposha* could be either a Sīhala commentary or a version of the *uttaravihāra-aṭṭhakathā* from the Abhayagiri monastery (Guruge 2005). However, Toshiichi Endo held an opposite opinion (Endo 2006). Ann Heirman also assumes that the Abhayagirivihāra connection is possible in the Chinese version, and that the translator was under many different kinds of influences (Heirman 2004). Thus, she is also cautious about coming to a conclusion as to the origins of the *Shanjianlü piposha*. According to her, giving a definite answer to the exact role that the Abhayagirivihāra tradition plays in the *Shanjianlü piposha* is extremely difficult because very little is known about the Abhayagirivihārins' viewpoints. Gudrun Pinte argued in her dissertation that the *Shanjianlü piposha* preserves an older layer of the *Samantapāsādikā*, which itself underwent changes and was elaborated at a date following its translation into Chinese in 489 A.D. (Pinte 2012, p. 532).

The abovementioned Japanese scholars, Nagai and Mizuno, attributed the differences between the *Shanjianlü piposha* and *Samantapāsādikā* to the influence of the *Dharmaguptaka-vinaya*, which was translated into Chinese as the *Sifen lü* (T.1428, the Four-part Vinaya) around 410 A.D. in Chang'an by Buddhayaśas 佛陀耶舍 (dates of birth and death unknown), who recited the text by heart while Zhu Fonian 竺佛念 (dates of birth and death unknown) rendered it into Chinese. This idea still survived in the English translation of the *Shanjianlü piposha* made by P.V. Bapat and A. Hirakawa (Bapat and Hirakawa 1970, pp. L–LIII). This assumption of Dharmaguptaka influence might result from the fact that the eminent Japanese monk Gyōnen 凝然 (1240–1321) classified the *Shanjianlü piposha* as a commentary on the *Dharmaguptaka-vinaya* in his *Risshū Kōyō* 律宗綱要 (*The Outline of the Vinaya School*). He states, 善見論釋四分律 *Zenken ron shaku Shibunritsu*, or in classical Chinese, *Shanjian lun shi Sifen lü* ("The *Shanjian lun* explains the *Sifen lü*") (Satō 1994, trans., p. 247). *Shanjian lun* is an alternative name for the *Shanjianlü piposha*. Ann Heirman further argues that the influence attributed by Bapat and Hirakawa to the *Dharmaguptaka-vinaya* is at some points wrong, and in other cases, could equally be ascribed to Sarvāstivādin influence or to any of the other *vinaya*s preserved in Chinese translation. She emphasizes the fact that, in the fifth century in South China, the *Sarvāstivādin-vinaya* was far more influential than the *Dharmaguptaka-vinaya* (Heirman 2004). Based on previous studies, Bhikkhu Ñāṇatusita has concluded that the *Shanjianlü piposha* is not a genuine translation of the *Samantapāsādikā* without any Chinese influence, nor is it an original Chinese composition. Instead, it is a Chinese Buddhist hybrid composition. It mainly consists of an abridged translation of the *Samantapāsādikā*, into which large passages from the *Suttavibhaṅga* and other unidentified texts were inserted, perhaps copied from earlier Chinese translations of these works, and it was occasionally adapted to fit the *Sifen lü*, popular in China, so that it was more of use to Chinese monastics (Ñāṇatusita 2014).

However, scholars such as Nagai, Mizuno, Heirman, and Pinte simply mention Gyōnen's classification to illustrate that this misinterpretation comes from the *Risshū*

*Kōyō*, leaving its origin underexamined. Bhikkhu Ñaṇatusita also ignored the source of Gyōnen's classification. Finding out exactly where Gyōnen's classification issued from has significant implications for our understanding of the history of the transmission of the *Shanjianlü piposha* in medieval China and Japan. I have published two articles on the relationship between the *Shanjianlü piposha* and the Vinaya school. According to my analysis, many elements from the *Sifen lü* were inserted into the translation of the *Shanjianlü piposha*. Therefore, the *Shanjianlü piposha* has been regarded as a commentary on the *Sifen lü* in Chinese Buddhism, with its Theravādin origin unknown to Chinese Buddhist monks. This misinterpretation could be traced back to Dingbin 定賓 (active in the Kaiyuan period (713–741), dates of birth and death unknown) of the Xiangbu Vinaya school 相部律宗. It was carried on by later vinaya masters of the Nanshan Vinaya school in the later Tang and Song dynasties, such as Jingxiao 景霄 (?–927), Yunkan 允堪 (?–1061), and Yuanzhao 元照 (1048–1116), and it further exerted an influence on Gyōnen's understanding of the relationship between the *Shanjianlü piposha* and the Dharmaguptaka school.[2] However, in these two articles, the patriarchal lineages of vinaya transmission in both the Pāli sources and Chinese translation are not mentioned.

Based on the previous research, I continue the study on Dingbin's and Gyōnen's narratives and interpretations on the patriarchal lineage of vinaya transmission in the *Shanjianlü piposha*. I further demonstrate how Chinese and Japanese vinaya masters interpret the relationship between the *Sifen lü* and the patriarchal lineage of vinaya transmission, starting with Upāli, in the *Shanjianlü piposha*.

In Book 2 of the *Shanjianlü piposha*, it is said,

"In the Jambudīpa (Skt. Jambudvīpa; Ch. Yanfuli 閻浮利), I shall tell the names of [vinaya masters] in due order: first Youboli 優波離 Upāli, second Duoxieju 馱寫拘 Dāsaka, third Xunaju 須那拘 Soṇaka, fourth Xijiapo 悉伽婆 Siggava, fifth Mujianlianzi Dixu 目揵連子帝須 Moggaliputta Tissa. These five masters handed down the *vinayapiṭaka* in succession in the Jambudīpa, without any interruption in the *vinayapiṭaka* up to the Third Buddhist Council. After the Third [Council], at the time of entering into *parinibbāna* (Skt. *parinirvāṇa*), Moggaliputta Tissa handed it over to his disciple Moshentuo 摩哂陀 Mahinda, the son of King Aśoka. Moshentuo brought the *vinayapiṭaka* into the Sīhaḷadīpa (Skt. Siṃhaladvīpa). At the moment of entering into *parinibbāna*, Moshentuo handed [the *vinayapiṭaka*] over to his disciple Alizha 阿栗吒 Ariṭṭha. Since then it has been handed down till today. One should know this. Now I will state the names of masters of ancient times. Five masters brought the *vinayapiṭaka* from the Jambudīpa to the Sīhaḷadīpa: first Moshentuo, second Yidiyu 一地臾 Iṭṭhiya, third Yudiyu 欝帝臾 Uttiya, fourth Canpolou 參婆樓 Sambala, fifth Batuosha 拔陀沙 Bhadda. These five masters had perfect wisdom and unhindered supernatural powers as well as three insights, and instructed disciples in the Sīhaḷadīpa respectively. Moshentuo, at the time of entering into *parinibbāna*, handed [the *vinayapiṭaka*] over to Alizha. Alizha handed it over to his disciple Dixudaduo 帝須達多 Tissadatta; Dixudaduo handed it over to his disciple Jialuoxumona 伽羅須末那 Kālasumana; Jialuoxumona handed it over to his disciple Dijiana 地伽那 Dīghanāmaka; Dijiana handed it over to his disciple Xumona 須末那 Dīghasumana; Xumona handed it over to his disciple Jialuoxumona 伽羅須末那 Kālasumana; Jialuoxumona handed it over to his disciple Tanwude 曇無德; Tanwude handed it over to his disciple Dixu 帝須 Tissa; Dixu handed it over to his disciple Tipo 提婆 Deva; Tipo handed it over to his disciple Xumona 須末那 Sumana; Xumona handed it over to his disciple Zhuannajia 專那伽 Cūlanāga; Zhuannajia handed it over to his disciple Tanwupoli 曇無婆離 Dhammapālita; Tanwupoli handed it over to his disciple Qimo 企摩 Khema; Qimo handed it over to his disciple Youbodixu 優波帝須 Upatissa; Youbodixu handed it over to his disciple Fapo 法叵 Puppha; Fapo handed it over to his disciple Apoye 阿婆耶 Cūlābhaya (?); Apoye handed it over to his disciple Tipo 提婆 Cūladeva (?); Tipo handed it over to his disciple Sipo 私婆 Sīva."

于閻浮利地，我當次第説名字：第一、優波離，第二、馱寫拘，第三、須那拘，第四、悉伽婆，第五、目揵連子帝須。此五法師于閻浮利地，以律藏次第相付，不令斷絶，乃至第三集律藏。從第三之後，目揵連子帝須臨涅槃，付弟子摩哂陀。摩哂陀是阿育王兒也，持律藏至師子國。摩哂陀臨涅槃，付弟子阿栗吒。從爾已來，更相傳授至於今日，應當知之。我今説往昔師名，從閻浮利地，五人持律藏至師子國：第一、名摩哂陀，第二、名一地臾，第三、名臂帝臾，第四、名參婆樓，第五、名拔陀沙。此五法師，智慧無比，神通無礙，得三達智，于師子國各教授弟子。摩哂陀臨涅槃，付弟子阿栗吒，阿栗吒付弟子帝須達多，帝須達多付弟子伽羅須末那，伽羅須末那付弟子地伽那，地伽那付須末那，須末那付伽羅須末那，伽羅須末那付曇無德，曇無德付帝須，帝須付提婆，提婆付須末那，須末那付專那伽，專那伽付曇無婆離，曇無婆離付企摩，企摩付優波帝須，優波帝須付法叵，法叵付阿婆耶，阿婆耶付提婆，提婆付私婆。[3]

Now, we move on to the parallel in the *Samantapāsādikā*:

"Jambudīpe tāva Upālittheram ādiṃ katvā ācariyaparamparāya yāva tatiyasaṅgī ti tāva ābhataṃ. tatrāyaṃ ācariyaparamparā:

Upāli Dāsako c'eva, Soṇako Siggavo tathā,

Tisso Moggaliputto ca, pañc'ete vijitāvino,

paramparāya vinayaṃ dīpe Jambusirivhaye

acchijjamānamānesuṃ, tatiyo yāva saṅgaho ti.

tass'attho ettavatā pakāsito hoti. tatiyasaṅgahato pana uddhaṃ imaṃ dīpaṃ Mahindâdīhi ābhataṃ. Mahindato uggahetvā kañci kālaṃ Ariṭṭhattherâdīhi ābhataṃ. tato yāva ajjatanā tesaṃ yeva antevāsikaparamparābhūtāya ācariyapara mparāya ābhatan ti veditabbaṃ. yathâhu porāṇā:

tato Mahindo Iṭṭiyo Uttiyo Sambalo pi ca

… … … … … … … [4] Bhaddanāmo ca paṇḍito;

ete nāgā mahāpaññā Jambudīpā idhâgatā:

vinayaṃ te vācayiṃsu piṭakaṃ Tambapaṇṇiyā.

nikāye pañca vācesuṃ satta c'eva pakāraṇe.

tato Ariṭṭho medhāvī Tissadatto ca paṇḍito

visārado Kālasumano, thero ca Dīghanāmako

… … … … … … … … … (see note 4) Dīghasumano ca paṇḍito.

punar eva Kālasumano Nāgatthero ca Buddharakkhito,

Tissatthero ca medhāvī Devatthero ca paṇḍito.

punar eva Sumano medhāvī vinaye ca visārado,

bahussuto Cūlanāgo, gajo 'va duppadhaṃsiyo.

Dhammapālitanāmo ca Rohaṇo sādhupūjito,

tassa sisso mahāpañño Khemanāmo tipeṭako.

dīpe tārakarājā 'va paññāya atirocatha,

Upatisso ca medhāvī Phussadevo mahākathī.

punar eva Sumano medhāvī, Pupphanāmo bahussuto,

mahākathī Mahāsivo piṭake sabbattha kovido.

punar eva Upāli medhāvī vinaye ca visārado,

mahānāgo mahāpañño, saddhammavaṃsakovido.

punar eva Abhayo medhāvī piṭake sabbattha kovido,

Tissatthero ca medhāvī vinaye ca visārado.

tassa sisso mahāpañño, Pupphanāmo bahussuto,

sāsanaṃ anurakkhanto Jambudīpe patiṭṭhito.

Cūlābhayo ca medhāvī vinaye ca visārado,

Tissatthero ca medhāvī saddhammavaṃsakovido.

Cūladevo ca medhāvī vinaye ca visārado

Sīvatthero ca medhāvī vinaye sabbattha kovido.

ete nāgā mahāpaññā vinayaññū maggakovidā,

vinayaṃ dīpe pakāsesuṃ piṭakaṃ Tambapaṇṇiyâti."[5]

It has been handed down in the Jambudīpa up to the Third Council by the succession of masters beginning with the Elder Upāli. Here is the succession of masters: Upāli, Dāsaka, as well as Soṇaka, similarly Siggava and Tissa Moggaliputta—these five victorious ones transmitted the vinaya in the glorious (is)land of Jambusiri (i.e., Jambudvīpa), in unbroken succession up to the time of the Third Council. And to this extent is its meaning declared. And after the time of the Third Council, it has been brought to this island by Mahinda and others. Having learned it from Mahinda, for some time, it was handed down by the Elder Ariṭṭha and others: and it should be known from that time up to the present day. It has been handed down by the succession of masters who constituted their own line of resident-pupils. According to the *porāṇa*s:

Thereupon Mahinda, Iṭṭhiya, Uttiya, Sambala and the learned Bhadda—these sinless sages of great wisdom came hither from Jambudīpa. They taught the *vinayapiṭaka* in the Tambapaṇṇi. They also taught five *nikāya*s and seven (*abhidhamma*) treatises. Then the wise Ariṭṭha and the learned Tissadatta, the skilled Kālasumana, the Elder named Dīghanāmaka and the learned Dīghasumana, and another Kālasumana, the Elder Nāga, Buddharakkhita, the wise Elder Tissa and the learned Elder Deva, and another wise Sumana proficient in the vinaya, Cūlanāga of great learning, unassailable as an elephant, and the Elder named Dhammapālita is like Mount Rohaṇa, revered by the virtuous. His pupil named Khema is of great wisdom and learned in three *piṭaka*s, who in his wisdom shone with great splendor in the island, like the king of the stars, Upatissa the wise, Phussadeva the great orator, and another wise Sumana, he of great learning named Phussa, the great orator Mahāsiva proficient in all the contents of the piṭaka, and again another wise Upāli skilled in the vinaya, Mahanāga of great wisdom, proficient in the tradition of the good teaching, and again the wise Abhaya skilled in all the contents of the *piṭaka*, the wise Elder Tissa proficient in the vinaya. His pupil named Puppha of great wisdom and of much learning, who while protecting the dispensation had established himself in the Jambudīpa. The wise Cūlābhaya proficient in the vinaya, the wise Elder Tissa skilled in the tradition of good teaching. Cūladeva the wise, proficient in the vinaya, and the wise Elder Sīva skilled in all the contents of the vinaya. These sinless sages of great wisdom, knowing the vinaya and skilled in the path, proclaimed the *vinayapiṭaka* on the island of the Tambapaṇṇi.[6]

This succession of vinaya masters found in the *Samantapāsādikā* is identical to the one from the *parivāra* in the Pāli *Vinaya*.[7] According to both the *Samantapāsādikā* and *Shanjianlü piposha*, the five masters, Upāli, Dāsaka, Soṇaka, Siggava, and Moggaliputta, transmitted the vinaya on the Indian continent. Mahinda, who was Tissa Moggaliputta's disciple, went to the island Tambapaṇṇi with Iṭṭhiya and three other masters to transmit the vinaya. There are thirty-three vinaya masters starting from Mahinda in this succession in the *Samantapāsādikā*, while there are only twenty-three masters in its parallel, the *Shanjianlü piposha*. K. Mizuno explained the differences between the two versions as some kind of confusion on the part of the Chinese translators because, for example, there were so many Tissas on the list. Besides this, he observed that the adjectives *dīgha*, *cūla*, and *nāma* have not always been transliterated or accurately translated (Mizuno 1996, p. 114). Gudrun Pinte assumed that Saṅghabhadra did not have a written document at hand, and that he remembered the material by heart, or rather most of it, or the co-translator Sengyi and his

team, who wrote the translation down in Chinese, simply became confused with lists of proper names (Pinte 2012, p. 50). It is difficult to give an exact answer about the reason why such differences arose between the two versions because so little is known to the translators. Based on Mizuno's research on the successions of vinaya masters in both versions, I drew up the following table to show the comparison of the lineages of vinaya masters (Table 1):

**Table 1.** A comparison of the lineages in the *Shanjianlü piposha* and Pāli sources.

| Lineage in Pāli Sources | Lineage in the *Shanjianlü piposha* |
| --- | --- |
| 1. Mahinda | 摩哂陀 |
| 2. Iṭṭiyo | 一地臾 |
| 3. Uttiyo | 欝帝臾 |
| 4. Sambalo | 參婆樓 |
| 5. Bhadda | 拔陀沙 |
| 6. Ariṭṭha | 阿栗吒 |
| 7. Tissadeva | 帝須達多 |
| 8. Kālāsumana | 伽羅須末那 |
| 9. Dīghanāmaka | 地伽那 |
| 10. Dīghasumana | 須末那 |
| 11. Kālasumana | 伽羅須末那 |
| 12. Nāga | |
| 13. Buddharakkhita | 曇無德 |
| 14. Tissa | 帝須 |
| 15. Deva | 提婆 |
| 16. Sumana | 須末那 |
| 17. Cūḷanāga | 專那伽 |
| 18. Dhammapālita | 曇無婆離 |
| 19. Khema | 企摩 |
| 20. Upatissa | 優婆帝須 |
| 21. Phussadeva | |
| 22. Sumana | |
| 23. Pupphanāma | |
| 24. Mahāsīva | |
| 25. Upāli | |
| 26. Mahānāga | |
| 27. Abhaya | |
| 28. Tissa | |
| 29. Pupphanāma | 法叵 |
| 30. Cūlābhaya | 阿婆那(?) |
| 31. Tissa | |
| 32. Cūladeva | 提婆(?) [1] |
| 33. Sīva | 私婆 |

[1]. According to Mizuno, Apona 阿婆那 is the Chinese transliteration of Cūlābhaya, and Tipo 提婆 is the Chinese transliteration of Cūladeva (Mizuno 1996, pp. 113–14). However, I am sceptical about this and thus add question marks.

As is shown in Table 1, Buddharakkhita, the 13th patriarch in the *Samantapāsādikā*, has Tanwude in the parallel of the *Shanjianlü piposha*. However, Buddharakkhita should be translated into Chinese as Fohu 佛護, and there is no Pāli name corresponding to Tanwude in the succession of vinaya masters in Pāli sources, including the Pāli *Vinaya*, *Samantapāsādikā*, *Mahāvaṃsa*, and *Dīpavaṃsa*. According to R. Saloman's study of a Gāndhārī inscription on a pot (Salomon 1999, p. 214), Tanwude is probably the transliteration of the Gāndhārī "dhamaüte" (Skt. dharmagupta; Pā. dhammagutta), which means Fazang 法藏 or Fahu 法護 in Chinese. Thus, the Dharmaguptaka school, which specifically promotes the *Sifen lü*, has Fazang bu 法藏部 or Tanwude bu 曇無德部 as its Chinese translation. The name Tanwude, which also appears in some other parts of the *Shanjianlü piposha*, is usually used as the translation of the Pāli term "Dhammarakkhita" rather than "Dhammagutta". However, the transliteration of "Dhammarakkhita" is Tanmo leqiduo 曇摩勒棄多, which also means Fahu in Chinese. Both "gutta" (Skt. gupta) and "rakkhita" mean protection in Pāli, and therefore, "Dhamma-gutta" and "Dhamma-rakkhita" are literally synonymous. As a result, translators of the *Shanjianlü piposha* chose the term "Tanwude" to translate its literally synonymous term, "Dhamma-rakkhita", as "Tanwude" usually appears in other Chinese Buddhist texts that predate the *Shanjianlü piposha* and thus is better known to Chinese readers. There are many similar cases in the *Shanjianlü piposha*. For instance, the Pāli term *nikāya* is not well known to Chinese Buddhist monks, so translators use the synonymous term *ahan* 阿含 (Skt. *āgama*) instead to paraphrase *nikāya*, which is already well known to Chinese readers.

K. Mizuno infers that the difference between Tanwude in the *Shanjianlü piposha* and Buddharakkhita in the *Samantapāsādikā* is a mistake caused by a certain reason, for which he gives no further explanation (Mizuno 1996, p. 114). Buddharakkhita is also mentioned in other chapters of the *Shanjianlü piposha* and *Samantapāsādikā*, and it is translated into Fowude 佛無德 or Fotuo leqiduo 佛陀勒棄多 in the *Shanjianlü piposha*. In Book 5 of the *Shanjianlü piposha*, it is said,

> "there are more than one kind of surnames, e.g., the surname of Gotama (Ch. Qutan 瞿曇), or the surname of Moggallāna (Ch. Mujianlian 目揵連), as well as more than one kind of given names, e.g., the given name of Buddharakkhita (Ch. Fowude 佛無德), or the given name of Dhammarakkhita (Ch. Tanwude 曇無德)".

> 姓非一種，名非一種，或姓瞿曇，或姓目揵連，或名佛無德，或名曇無德。[8]

Here, the translation "Fowude" is an imitation of "Tanwude", both of which serve as examples to explain Indians' given names rather than certain individuals. In Book 10 of the *Shanjianlü piposha*, it is stated,

> "One gives a verbal command to another" means: There are a number of *bhikkhu*s. One of them is a teacher, and the other three are pupils. The first pupil's name is Buddharakkhita, the second is Dhammarakkhita, and the third is Saṅgharakkhita. The teacher sees an object belonging to others, and the thought of stealing it arises in his mind. He calls Buddharakkhita with these words: 'You command Dhammarakkhita to instruct Saṅgharakkhita in going to take that object away.' At the very moment he commands the first pupil, the teacher becomes guilty of *dukkaṭa*. When Dhammarakkhita instructs [Saṅgharakkhita] and when Saṅgharakkhita receives the instruction, the teacher becomes guilty of *thullaccaya*. If [the third pupil] removes the object from its original place, the teacher and his three pupils all become guilty of a grave offense."

> 教語此人者，有衆多比丘，一是師、三是弟子，第一弟子名佛陀勒棄多（Pā. Buddharakkhita），二名曇摩勒棄多（Pā. Dhammarakkhita），三名僧伽勒棄多（Pā. Saṅgharakkhita）。師行見他物，起盜心，喚佛陀勒棄多語言：" 汝教曇摩勒棄多，教僧伽勒棄多，往取彼物。" 師教第一弟子時，師得突吉羅。曇摩勒棄多語、僧伽勒棄多受語時，師得偷蘭遮。若往取物離本處，師及三弟子俱犯重。[9]

This passage corresponds to its parallel in the *Samantapāsādikā* and explains the details of the law of theft with the case of four *bhikkhu*s. It is clear that the names of Buddharakkhita and Dhammarakkhita mentioned here have nothing to do with those in the succession of vinaya masters. We can see that a proper name usually has various translations or transliterations in the *Shanjianlü piposha*. For instance, the name of *Visuddhimagga* is mentioned at least three times in the *Shanjianlü piposha* and each has a different translation: *Jingdao jing* 淨道經, *Jingdao piposha* 淨道毗婆沙, and *Apitan piposha* 阿毗曇毗婆沙. The *Shanjianlü piposha* was not translated by a well-organized translation team with a highly specialized division of labor, nor with adequate proofreading. Consequently, there are many inconsistencies in the Chinese translation. However, generally speaking, both the terms *buddha* and *dhamma* have specific meanings in Buddhism, which could hardly be confused by the translators. Regarding the difference between Buddharakkhita and Tanwude, it appears to be a deliberate change made by later people because the *Dharmaguptaka-vinaya* had not yet achieved a dominant position in China around 488 and 489 A.D. It is very difficult to figure out why Buddharakkhita was "translated into" Tanwude in the absence of crucial historical evidence. However, I think, as far as this "mistranslation" is concerned, two questions should be focused on: Firstly, how does this "mistranslation" in the *Shanjianlü piposha* exert influence on Chinese and Japanese Buddhist monks' identification of the Indian origin of the Dharmaguptaka school (Ch. Fazang bu 法藏部)? Secondly, how do Chinese and Japanese Buddhist monks interpret this patriarchal lineage of vinaya transmission in the *Shanjianlü piposha* in their Vinaya school works?

### 2.2. The Interpretations Made by the Vinaya School in China and Japan

The Vinaya school is a scholastic tradition of East Asian Buddhism based on the study of the *Dharmaguptaka-vinaya*. There are three branches of the Vinaya school: the Nanshan Vinaya school associated with Daoxuan 道宣 (596–667), the Xiangbu Vinaya school associated with Fali 法礪 (569–635), and the Dongta Vinaya school 東塔律宗 associated with Huaisu 懷素 (625–698). Of these three, the Nanshan Vinaya school eventually eclipsed the other two. Monks from both the Xiangbu and Nanshan Vinaya schools had their own Sinicized interpretations of the patriarchal lineage of vinaya transmission in the *Shanjianlü piposha*, while monks from the Dongta Vinaya school paid little attention to this lineage, according to the historical records we see today.

### 2.2.1. Dingbin's Interpretation

The first person who notices the patriarchal lineage of vinaya transmission in the *Shanjianlü piposha* is Dingbin, who is a vinaya master of the Xiangbu Vinaya school in the Kaiyuan 開元 period of the Tang dynasty. Dingbin wrote a subcommentary on Fali's *Sifen Lü* commentary called *Sifen lüshu shi zong yi ji* 四分律疏飾宗義記 (*For the Decoration of the School: Study on [Fali's] Sifen Lü Commentary*) that is signed as a *śramaṇa* in the Zhenguo *Bodhimaṇḍa* in Mount Song (Songyue zhenguo daochang shamen 嵩岳鎮國道場沙門), and therefore he is also known as the vinaya master of Songyue 嵩岳律師.[10] In his subcommentary, he extensively quotes the stories about the Third Council and King Aśoka's mission to spread Buddhism that are recorded in the *Shanjianlü piposha*:

> During the Third Council, two sects have already formed. However, in this commentary (i.e., the *Shanjianlü piposha*), it is argued that there exists only one sect that has been handed down. Consequently, the distinguishing characteristics of split sects are ignored in this commentary. From that time onwards, [Tissa Moggaliputta 目揵連子帝須] handed the *vinayapiṭaka* over to Moshentuo, the son of King Aśoka. Moshentuo handed the *vinayapiṭaka* over to Alizha, and Alizha handed the *vinayapiṭaka* over to his disciple Dixudaduo. The next successor is Jialuoxumona . . . The next successor is Sipo, twenty-four masters in total[11] . . . According to the *Shanjianlü piposha*, the thirteenth patriarch in these twenty-four is named Tanwude. I read through this commentary from beginning to end and find that it shares a very similar structure with the *Sifen lü*, and many passages

in both are corresponding. Therefore, this *Shanjianlü piposha* is a commentary on the *Sifen lü*. And this Tanwude is the master of this *Sifen lü*. It is also said in this commentary that the master Mohe tanwude 摩訶曇無德 went to the Abo lanruo guo 阿波蘭多國 (Pā. Aparāntaka) for the purpose of transmitting the *vinayapiṭaka*. Here this [Mohe tanwude] is not the name of the master of *Sifen lü*. For, in no context is this Mohe tanwude considered to be the name of the master of *Sifen lü*. It is asked: as the former Tanwude is considered as a religious name (Ch. faming 法名) of a Buddhist monk, why is this [Mohe tanwude] stated to be a secular personal name? The answer is: a master is named after the *dharma*s he transmits. For instance, masters [who transmit Chan *dharma*s] are nowadays called Chan masters (Ch. chanshi 禪師) etc."

第三結集之時，因分二部，然由此論，但欲自辨一支相傳，故略不説分部差別也。從此已後，(目揵連子帝須) 付摩哂陀，此即育王之子也。摩哂陀付阿栗吒，阿栗吒付弟子帝須達多，次伽羅須末那 ...... 次私婆，合二十四人。 ...... 又准《見論》，二十四人之中，第十三人，名曇無德者。竊尋彼論，勘其始末，其與《四分》科段相當，故知彼論釋《四分律》。其曇無德即是此律主也。彼論複説，摩訶曇無德，至阿波蘭多國，流通律藏。此即非是律主名也，以其無文云是摩訶曇無德故也。問前已成立曇無德者，乃是法名。何故今言是人名也？答：此蓋就所弘法，以號其人，如即今人號禪師等。[12]

The quoted passage reveals that Dingbin's interpretation is deeply influenced by the "mistranslated" Tanwude in the succession of vinaya masters in the *Shanjianlü piposha*. Firstly, the earliest Chinese Buddhist work in which Tanwude is considered to be the master of *Sifen lü* is *Lidai sanbao ji* 歷代三寶紀 (*Records of Three Treasures Through the Ages*) was written by Fei Changfang 費長房 (dates of birth and death unknown) in the Sui dynasty.[13] It could be inferred that Dingbin identified the master of *Sifen lü* as this thirteenth patriarch, Tanwude, in the lineage of vinaya transmission based on his own standpoint towards the Xiangbu Vinaya school. Secondly, Dingbin also regards the master Mohe tanwude 摩訶曇無德 (Pā. Mahādhammarakkhita) in Aśoka's mission as the promoter of the *Dharmaguptaka-vinaya*. The name Mohe tanwude is formed by "mohe 摩訶" and "Tanwude 曇無德". Tanwude is the historical figure who compiled the *Dharmaguptaka-vinaya* and founded the Dharmaguptaka school, known as Tanwude bu in Chinese. Therefore, according to Dingbin, a person is named Mohe tanwude due to his promotion of the Dharmaguptaka doctrine, just as nowadays monks who promote Chan Buddhism are called Chan masters.

However, regarding Mahādhammarakkhita or Mohe tanwude in the *Samantapāsādikā* and *Shanjianlü piposha*, we cannot find any definite Dharmaguptaka connection. In Aśoka's mission, the master Yonaka-Dhammarakkhita, who came from the Yonaka (Ch.Yuna guo 臾那國), was sent to preach *Aggikkhandhopama* (Ch. *Huoju piyujing* 火聚譬喻經) for the purpose of spreading Buddhism in the Aparantaka (Ch. Abo lanruo guo 阿波蘭多國), and the master Mahādhammarakkhita was sent to preach the *Mahānāradakassapajātaka* (Ch. *Mohe naluotuo jiaye benshengjing* 摩訶那羅陀迦葉本生經) in order to spread Buddhism in the Mahāraṭṭha (Ch. Mohe lezha guo 摩訶勒吒國).[14] Neither *Aggikkhandhopama*,[15] which is found in the present *Aṅguttaranikāya*, nor *Mahānāradakassapajātaka*,[16] which is found in the present *Khuddakanikāya*, can be attributed to the Dharmaguptaka school. Erich Frauwallner proposes that both Dhammarakkhitas in the mission are related to the origin of the Dharmaguptaka school, but no historical evidence is presented in his hypothesis (Frauwallner 1956, p. 22). There is no evidence to confirm a definite connection between this Mahādhammarakkhita and the Dharmaguptaka school. Thus, it is a fact that Dingbin distorted the meaning of the context by quoting fragments from passages in the *Shanjianlü piposha*. As told by him, the *Shanjianlü piposha* and *Sifen lü* share a similar structure, and many passages in both texts are corresponding 其與《四分》科段相當. As a result, he misidentified the *Shanjianlü piposha* as a commentary on the *Sifen lü* due to his lack of learning on the Theravādin vinaya. According to the historical records we have today, Dingbin is the first one to misunderstand the *Shanjianlü piposha* as a commentary on the

*Sifen lü*. That is to say, no later than the Kaiyuan period in the Tang dynasty, the school affiliation of the *Shanjianlü piposha* had been interpreted as the Dharmaguptaka by the Chinese Vinaya school.

Dingbin noticed the correspondence between the *Shanjianlü piposha* and *Sifen lü*, but he presented no detailed discussion. M. Nagai and K. Mizuno have performed comparative studies on both texts. M. Nagai points out that the ordering of the 85th–91st *pācittiya*s and some *Khandhaka* (Ch. qiandu 犍度) chapters in the *Shanjianlü piposha* are consistent with those in the *Sifen lü*. Table 2 shows the comparison of relevant references in the *Sifen lü*, *Shanjianlü piposha*, and *Samantapāsādikā*.

**Table 2.** A comparison of the ordering of the 85th–91st *pācittiya*s and *khandhaka*s in the *Sifen lü*, *Shanjianlü piposha*, and *Samantapāsādikā*.

| The *Sifen lü* | The *Shanjianlü piposha* | The *Samantapāsādikā* |
|---|---|---|
| From the 85th to 91st *pācittiyas* | | |
| 非時入聚落戒 rules for entering a village out of hours | 非時入聚落戒 rules for entering a village out of hours | *vikālagāmappavisana-sikkhāpada* (rules for entering a village out of hours) |
| 過量床足戒 rules for excessive feet of bedsteads and chairs | 高床戒 rules for bedsteads and chairs | *sūcighara-sikkhāpada* (rules for needle cases) |
| 兜羅貯床褥戒 rules for bedsteads stuffed with cotton | 兜羅綖坐褥戒 rules for chairs stuffed with cotton | *mañca-sikkhāpada* (rules for bedsteads and chairs) |
| 骨牙角 作針筒戒 rules for needle cases made of bones, teeth and horns | 針筒戒 rules for needle cases | *tūlonaddha-sikkhāpada* (rules for bedsteads and chairs stuffed with cotton) |
| 過量尼師檀戒 rules for excessive mats | 尼師檀戒 rules for mats | *nisīdana-sikkhāpada* (rules for mats) |
| 覆瘡衣過量戒 rules for excessive garments for covering sores | 覆瘡衣戒 rules for garments for covering sores | *kaṇḍupaṭicchādi-sikkhāpada* (rules for garments for covering sores) |
| 雨衣過量戒 rules for excessive garments made for the rainy season | 雨浴衣戒 rules for garments made for the rainy season | *vassikasāṭika-sikkhāpada* (rules for garments made for the rainy season) |
| Khandhakas | | |
| 1 受戒犍度 on the ordination of Buddhist monks | 1 受戒犍度 on the ordination of Buddhist monks | 1. *mahākhandhaka* (the great section) [1] |
| 2 説戒犍度 [2] on teaching the precepts | 2 布薩犍度 on the *uposatha* | 2. *uposathakkhandhaka* (on the *uposatha*) |
| 3 安居犍度 on the rains | 3 安居犍度 on the rains | 3. *vassūpanāyikakkhandhaka* (on the rains) |
| 4 自恣犍度 on teachings regarding self-indulgence | 4 皮革犍度 on the use of leather | 4. *pavāraṇākkhandhaka* (on teachings regarding self-indulgence) |
| 5 皮革犍度 on the use of leather | 5 衣犍度 on robes | 5. *cammakkhandhaka* (on the use of leather) |
| 6 衣犍度 on robes | 6 藥犍度 on medicines | 6. *bhesajjakkhandhaka* (on medicines) |
| 7 藥犍度 on medicines | 7 迦絺那衣犍度 on the *kathina* | 7. *kathinakkhandhaka* (on the *kathina*) |
| 8 迦絺那衣犍度 on the *kathina* | 8 別住犍度 on isolation for improper conduct | 8. *cīvarakkhandhaka* (on robes) |
| 9 拘睒彌犍度 on [monks] at *Kosambī* | 9 拘睒彌犍度 on [monks] at *Kosambī* | 9. *campeyyakkhandhaka* (on [monks] at *Campā*) |
| 10 瞻波犍度 on [monks] at *Campā* | 10 瞻波犍度 on [monks] at *Campā* | 10. *kosambakakkhandhaka* (on [monks] at *Kosambī*) |
| 11 呵責犍度 on rebuking quarrelsome monks | 11 滅諍犍度 on resolution of disputes | 11. *kammakkhandhaka* (on formal acts) |
| 12 人犍度 [3] on correction of minor crimes | 12 比丘尼犍度 on Buddhist nuns | 12. *pārivāsikakkhandhaka* (on isolation for improper conduct) |

**Table 2.** *Cont.*

| The *Sifen lü* | The *Shanjianlü piposha* | The *Samantapāsādikā* |
|---|---|---|
| 13 覆藏揵度 on remedies for those who conceal their crimes | 13 法揵度 on ritual performances [4] | 13. *samuccayakkhandhaka* (on accumulation of [offences]) |
| 14 遮揵度 on dealing with offenses not treated at the *uposatha* | | 14. *samathakkhandhaka* (on settlements of legal questions) |
| 15 破僧揵度 on destruction of the *saṅgha* | | 15. *khuddakavatthukkhandhaka* (on minor matters) |
| 16 滅諍揵度 on resolution of disputes | | 16. *senāsanakkhandhaka* (on lodgings) |
| 17 比丘尼揵度 on Buddhist nuns | | 17. *saṅghabhedakakkhandhaka* (on destruction of the *saṅgha*) |
| 18 法揵度 on ritual performances | | 18. *vattakkhandhaka* (on observances) |
| 19 房舍揵度 on lodgings | | 19. *pātimokkhaṭṭhapanakkhandhaka* (on suspending the *pātimokkha*) |
| 20 雜揵度 on miscellany | | 20. *bhikkhunikkhandhaka* (on Buddhist nuns) |
| | | 21. *pañcasatikakkhandhaka* (on the Five Hundred) |
| | | 22. *sattasatikakkhandhaka* (on the Seven Hundred) |

[1] "The mahākhandhaka" in Pāli sources deals with the ordination of Buddhist monks, which is equivalent to 受戒揵度. [2] The 說戒揵度 in the *Sifen lü* deals with *uposatha* ceremony, which is equivalent to 布薩揵度. [3] The 人揵度 in the *Sifen lü* deals with isolation for monks who are guilty of *saṃghāvaśeṣa*, which is equivalent to 別住揵度. [4] The editors of the *Taishō* version interpret 法揵度 as the Chinese translation for the Pāli *vattakhandhaka*. However, in fact, the 法揵度 in the *Shanjianlü piposha* deals with lodgings, which has its parallel in the *senāsanakkhandhaka* chapter in the *Samantapāsādikā*. In Pāli sources, the *vattakhandhaka* chapter deals with Buddhist monks' manners and behaviors. The term 法揵度 only appears in the *Sifen lü* and *Shanjianlü piposha*. In both texts, the chapters before 法揵度 are 比丘尼揵度.

As is seen in Table 2, on the one hand, the *khandhaka* part of the *Shanjianlü piposha* is much shorter than that of the *Samantapāsādikā*. On the other hand, except for the *Biezhu qiandu* 別住揵度, the order of the *khandhaka* chapters in the *Shanjianlü piposha* is nearly the same as that in the *Sifen lü*. According to K. Mizuno, the *Yao qiandu* 藥揵度 (the *khandhaka* chapter on medicines) and *Pige qiandu* 皮革揵度 (the *khandhaka* chapter on the use of leather) not only share the same order in both the *Shanjianlü piposha* and *Sifen lü*, but they also have the same textual content, which indicates the definite influence of the Dharmaguptaka school on the *Shanjianlü piposha* (Mizuno 1996, pp. 89–96). Apart from this, in the *Shanjianlü piposha*, there are precepts about the *stūpa* directly copied from the *Sifen lü*.

The two precepts about staying overnight in or hiding one's things in a shrine of the *stūpa* of the Buddha did not exist in the original Indic text. They did not exist because when the Buddha was alive, there could not have been any *stūpa* of his. These precepts (in the *prātimokṣa*) were laid down by the Buddha: [It is not allowed to] enter the *stūpa* with leather-shoes on, or when one holds them in his hand; [It is not allowed to] enter the *stūpa* of the Buddha with a leg-cover-shoe on, or when one holds it in his hand; [It is not allowed] to eat at the foot of the *stūpa* of the Buddha, or to carry a dead body on one's shoulders and burn it at the foot of the *stūpa* of the Buddha, or to burn it in front of the *stūpa*, or to burn the dead body around on any of the four sides of the *stūpa*. So also, one is not permitted to carry the clothes or a bed-cot of a dead person across the foot of the *stūpa*. One is not permitted to answer the calls of nature at the foot of a *stūpa*, nor in front of it, nor around the *stūpa* of the Buddha. One is not permitted to approach the place for answering the calls of nature while holding a Buddha image in his hand. One is not permitted to bite and chew a tooth-stick at the foot of a *stūpa* of the Buddha, nor in front of it, nor around any of its four sides. One is not permitted to drop mucus [from his nose], or saliva [from his mouth] at the foot of a *stūpa* of the Buddha, or in its front, or any of the four sides. One is not permitted

to stretch his legs towards a *stūpa* of the Buddha; nor can one place the Buddha image in a room on a lower level. These precepts, more than twenty, did not exist in the original Indic text, as the Buddha was alive and, hence, no *stūpa* existed.

> 佛塔中止宿及藏物，此二戒梵本無有。所以無者，佛在世未有塔。此戒佛在世制。是故無著革屣入佛塔，手捉革屣入佛塔，著腹羅入佛塔，手捉腹羅入佛塔，佛塔下食擔死尸，塔下燒死尸，向塔燒死尸，繞塔四邊燒死尸，不得擔死人衣及床從塔下過，佛塔下大小便，向佛塔大小便，繞佛塔大小便，不得持佛像至大小便處，不得佛塔下嚼楊枝，不得向佛塔嚼楊枝，不得繞佛塔四邊嚼楊枝，不得佛塔下涕唾，不得向佛塔涕唾，不得繞佛塔四邊涕唾，向佛塔舒脚，安佛置下房。此上二十戒，梵本無有，如來在世塔無佛故。[17]

K. Mizuno noticed this passage and found out that the *prātimokṣa* of the *Sifen lü* gives rule nos. 60–85, dealing with the *stūpa* or image of the Buddha, to which this passage closely corresponds.[18] To sum up, the correspondence to the *Sifen lü* mainly lies in the latter part (i.e., some *pācittiya* rules and *khandhaka* chapters) of the *Shanjianlü piposha*. Though the corresponding part does not make up a major percentage of the total text, it shows a clear indication of the Dharmaguptaka connection.

### 2.2.2. Gyōnen's Interpretation

Dingbin's work spread to Japan and deeply influenced Japanese Buddhism after Jianzhen 鑒真 (Jp. Ganjin, 688–763) crossed over to Japan in the Tianbao 天寶 period of the Tang dynasty (X. Wang 1979, annotated, pp. 88–96). Influenced by Dingbin, Gyōnen, an eminent Japanese monk learned in doctrines of both the Xiangbu and Nanshan Vinaya schools, also misinterpreted the *Shanjianlü piposha* as a commentary on the *Sifen lü* in his *Risshū Kōyō* (Satō 1994, trans., p. 247), where he quoted Dingbin's abovementioned passage and gave further analysis as follows:

> In the *Shanjian* 善見 (i.e., *Shanjianlü piposha*), ancient masters are listed. However, the chronology of these masters is not mentioned. It is said in this commentary (i.e., *Shanjianlü piposha*), by the time the elders arrived in the Siṃhaladvīpa, with Moshentuo as the head master, 236 years after the Buddha's *nirvāṇa* had passed. When Buddhist doctrines were transmitted to the Siṃhaladvīpa, Moshentuo who was the sixth patriarch in the lineage of vinaya masters, had been transmitting and holding Buddhist doctrines at that time. The Tanwude, who is the thirteenth patriarch in the lineage of vinaya masters, is identified by Dingbin as the master of this *Vinaya* (i.e., the *Sifen lü*). Today it is clearly known that the Tanwude, the master of *Sifen lü*, lived around one hundred years after the Buddha's *nirvāṇa*. However, according to the *Shanjian lun* 善見論 (i.e., the *Shanjianlü piposha*), Moshentuo, the sixth patriarch in the lineage, lived more than two hundred years after the Buddha's *nirvāṇa*. So if [the date of] this Tanwude, the thirteenth patriarch, [is ascribed to around one hundred years after the Buddha's *nirvāṇa*], is it matchable [or reasonable]? [Of course, it is not the case.] Therefore, it should be inferred that in the twenty schools of Buddhism, the Dharmaguptaka school is also known as Fazang bu 法藏部, Fami bu 法密部, Fahu bu 法護部, and Fazheng bu 法正部, which emerged 380 years [after the Buddha's *nirvāṇa*].[19] This date could match the chronological record in the *Jian lun* 見論 (i.e., the *Shanjianlü piposha*). According to Dingbin, the master of *Sifen lü*, who lived around one hundred years after the Buddha's *nirvāṇa*, had the same name with the founder of the Dharmaguptaka school. Therefore, this founder is also considered as the master of *Sifen lü* [due to promotion of the *Sifen lü* by the Dharmaguptaka school]. Isn't there any contradiction in this statement?[20]

> 《善見》列諸師，未別指時代。然彼論云，爾時，諸大德到師子州中已，摩哂陀爲上座，于時佛涅槃已二百三十六歲。佛法通流至師子州中，哂陀即是第六傳律，乃在彼時，傳持佛法。彼第十三曇無德者，嵩岳定賓律師判云，其曇無德即是此律主也。今詳，《四分》律主曇無德者，如來滅後百年時出，《善見論》



意，第六摩哂陀既是二百餘年而出，況第十三豈相符乎？是故應言二十部中，曇
無德部，此云法藏，亦云法密，亦云法護，亦云法正。法藏三百八十年起，與
《見論》意時分相稱。嵩岳師意，彼興百年時《四分》律主其名既同，故後法藏
言此律主，有何遮妨？

The quoted passage indicates that Gyōnen agreed with Dingbin and had his own further understanding. Firstly, he states that Tanwude, the master of *Sifen lü*, lived around one hundred years after the Buddha's *nirvāṇa*, which is correspondent to the Chinese vinaya master Zhihong's 志鴻 (alive in the Tang dynasty, dates of birth and death unknown) saying in his *Sifen lü xingshichao sou xuan lu* 四分律行事鈔搜玄錄 (*Investigation: Study on [Daoxuan's Xingshi Chao]*):

> "Four-part" means: according to the *Fufazang zhuan* 付法藏傳 (i.e., *Fufazang yin yuan zhuan* 付法藏因緣傳), 優波毱多 (Skt. Upagupta) had five disciples. After one hundred years after the Buddha's *nirvāṇa*, each of them believed in their own claims for the vinaya which were taken as their own guidelines, and hereby the basic vinaya was divided into five sects of classics. The proper name "four-part" thus emerged. As ancient masters said, a vinaya master named Tanwude, four times edited and transmitted the great [Vinaya]*piṭaka* in Eighty Recitations 大藏八十誦律, with full annotations and interpretations. Therefore, [the vinaya edited and transmitted by Tanwude] is named "the Four-part [Vinaya]".

> 言四分者，《付法藏傳》云，百年之後，優波毱多有五弟子，各執一見，以爲揩准，遂分大藏，以爲五典。四分別號，從此而興，古師云，曇無德律主，於大藏八十誦律中四度傳文，盡所詮相，故云四分。[21]

As is recorded in the *Fufazang yin yuan zhuan* 付法藏因緣傳 (*The Work Explaining The Handing Down of Śākyamuni's Teaching by Mahākāśyapa and The Olders*), Upagupta, who lived around one hundred years after the Buddha's *nirvāṇa*, was predicted by the Buddha to be the one enriching all sentient beings.[22] Tanwude was Upagupta's disciple, both of whom lived in the same period. However, as is said in the *Shanjianlü piposha*, at the time when the elders arrived in the Siṃhaladvīpa with Mahinda as their leading master, it was 236 years since the Buddha's *nirvāṇa*. 諸大德到師子洲中已，摩哂陀爲上座。于時佛涅槃已二百三十六歲。[23] That is to say, Mahinda, the sixth patriarch in the lineage, lived more than two hundred years after the Buddha's *nirvāṇa*. In this case, how could Tanwude, the thirteenth patriarch in the lineage, have lived around one hundred years after the Buddha's *nirvāṇa*? Gyōnen's answer is as follows: The Dharmaguptaka school emerged 380 years after the Buddha's *nirvāṇa*. What Dingbin really meant is that this Tanwude, the thirteenth patriarch in the lineage in the *Shanjianlü piposha*, should be referred to as the founder of the Dharmaguptaka school that emerged in later times. Although he was also called Tanwude and shared the same name with the master of *Sifen lü* who lived around one hundred years after the Buddha's *nirvāṇa*, this Tanwude in the lineage was referred to as ci lüzhu 此律主 (the master of this *Vinaya* (i.e., the master of *Sifen lü*)) as well because the Dharmaguptaka school promoted the *Sifen lü*. In Gyōnen's interpretation above, the master of *Sifen lü* called Tanwude who lived around one hundred years after the Buddha's *nirvāṇa* was not the founder of the Dharmaguptaka school of the sectarian period. The Dharmaguptaka school, which specifically transmitted and promoted the *Sifen lü*, emerged more than two hundred years after the edition and compilation of the *Sifen lü*. Thus, the founder of the Dharmaguptaka school was also named Tanwude. Mahinda, the sixth patriarch in the lineage, lived 236 years after the Buddha's *nirvāṇa*. Tanwude, the thirteenth patriarch in the lineage, who should be the founder of the Dharmaguptaka school, lived around 380 years after the Buddha's *nirvāṇa*, much later than Mahinda. As Gyōnen finally concluded, the identification of this Tanwude in the lineage as the founder of the Dharmaguptaka school is matchable with the chronological record in the *Shanjianlü piposha* 與《見論》意時分相稱. We can conclude that, in order to solve the possible chronological problem in Dingbin's narrative, Gyōnen thought of a seemingly reasonable explanation to justify Dingbin's lineage assertion.

*2.3. The Patriarchal Lineage of Vinaya Transmission in the Mohe sengqi lü 摩訶僧祇律*

In the *Mohe sengqi lü* ascribed to the Mahāsāṃghika school, a Chinese vinaya text trans­lated by Faxian 法顯 (337–422) in the Eastern Jin dynasty, there exists another patriarchal lineage of vinaya transmission, as follows: Youboli 優波離 (Upāli)→Tuosuopoluo 陀娑婆羅→Shutiposuo 樹提陀娑→Qiduo 耆哆→Genhu 根護→Fagao 法高→Juxi 巨醯→Muduo 目哆→Nenghu 能護→Mohena 摩訶那→Moqiuduo 摩求哆→Jusheluo 巨舍羅→Niuhu 牛護→Shanhu 善護→Huming 護命→Chatuo 差陀→Yeshe 耶舍→Futiluo 弗提羅→Qipojia 耆婆伽→Fahu 法護→Tinajia 提那伽→Faqian 法錢→Longjue 龍覺→Fasheng 法勝→Sengjiatipo 僧伽提婆→Fushapotuoluo 弗沙婆陀羅→Daoli 道力.[24] There are twenty-seven masters in this lineage. However, their dates are not mentioned at all by the translator. This lineage also has Upāli as its first patriarch. There is a master named Fahu in it, who is interpreted by Gyōnen as follows:

> In this vinaya text, although twenty-seven masters are listed, it is not known how many years after the Buddha's *nirvāṇa* they lived. The twentieth master named Fahu shared the same name as the master of *Sifen lü*. In the Root Section 根本部, the master of *Sifen lü* is Tanwude, who lived around one hundred years after the Buddha's *nirvāṇa*. In the twenty schools, there is a Dharmaguptaka school, the founder of which had the same name as his predecessor but kept the root text *Mohe sengqi* 根本摩訶僧祇.[25] His date is 380 years [after the Buddha's *nirvāṇa*]. Isn't there any contradiction? Though [the founder] had his own school affiliation, he preached both [the *Mohe sengqi lü* and the *Sifen lü*].

> 彼律雖列二十七人，不明佛滅經幾許年。第二十師名曰法護，與《四分》律主名全同。而是根本部《四分》律主是百年時。二十部中有法藏部，彼部主取前人法名而持根本《摩訶僧祇》，在其三百八十年時，有何遮妨？雖有自計，兼弘爾故。(Satō 1994, trans., p. 232)

The dates of these twenty-seven vinaya masters are not clear due to a lack of historical evidence. According to Gyōnen, the name of the twentieth master (i.e., Fahu) and that of the master of *Sifen lü* (i.e., Tanwude) from a Root Section are literally synonymous.[26] The founder of the Dharmaguptaka school also took the name Tanwude, while he kept the *Mohe sengqi lü* as well. As Gyōnen thought, there was no problem for the founder of the Dharmaguptaka school to preach both the *Mohe sengqi lü* and *Sifen lü*, despite his own school affiliation. With the aim of asserting that the Vinaya school had a direct lineage from Indian patriarchs beginning with Upāli, Gyōnen made an artificial link between this Fahu in the *Mohe sengqi lü* and the promotion of the *Sifen lü*, assuming that the transmission of the *Mohe sengqi lü* was also linked to the Dharmaguptaka school.

## 3. Construction and Critique of Patriarchal Lineages of Vinaya School Starting with Upāli

### 3.1. Construction of Patriarchal Lineages of Vinaya School Starting with Upāli

Besides Dingbin's efforts to claim a direct lineage from Upāli, the discussion on the origin of the Vinaya school continued in later periods. The Xiangbu Vinaya school declined and gradually merged into the Nanshan Vinaya school after Dingbin (J. Wang 2008, p. 259). In the Song dynasty, the construction of a patriarchal genealogy of the Vinaya school was a prevailing practice among eminent monks for the purpose of inheriting and developing the Nanshan Vinaya school.

The vinaya master Puning 普寧 established five patriarchs:

Upāli→Fazheng 法正 (i.e., Dharmagupta)→Jueming 覺明 (i.e., 佛陀耶舍 Buddhayaśas, the translator of the *Sifen lü*)→Zhishou→Nanshan 南山 (i.e., Daoxuan).

Renyue 仁嶽 established ten patriarchs:

Upāli→Fazheng→Jueming→Facong 法聰→Daofu 道覆→Huiguang 慧光→Daoyun 道雲→Daohong 道洪→Zhishou→Daoxuan.

Shouren 守仁 established seven patriarchs:

Upāli→ Fazheng→Jueming→Facong→Zhishou→Daoxuan→the authors of the *Zenghuiji* 增輝記主.[27]

Likewise, Renkan 仁堪 established seven patriarchs:

Upāli→Fazheng→Tandi 曇諦→Jueming→Facong→ Zhishou→Daoxuan.[28]

These four masters from the Nanshan Vinaya school claimed that the vinaya canon was handed down directly from Upāli to Fazheng, the founder of the Dharmaguptaka school. But the accurate dates of both Upāli and Dharmagupta are obscure. Nothing seems to have predestined Dharmagupta to become the successor to Upāli. In this case, the order of the basic succession—from Upāli to Dharmagupta—was called into question and severely criticized by Yuanzhao, an eminent monk from the Nanshan Vinaya school of the same period.

*3.2. Yuanzhao's Criticism*

Yuanzhao opposed such a construction of patriarchal genealogies going back to Upāli in the *Zhiyuan yibian* 芝園遺編 (*The Collected Posthumous Works of Yuanzhao*), edited by his disciple Daoxun 道询:

> Upāli was identified as the first patriarch by these four masters. However, there are three reasons for such an untenable lineage assertion. Firstly, the fundamental *vinayapiṭaka* compiled and recited by Upāli is the present *Mohe sengqi lü* ascribed to a Root Section. Although the [Dharmaguptaka] school which the *Sifen lü* is ascribed to have derived from this [Root Section], the fundamental sects and their branches co-existed and competed with each other, starting in the sectarian period. As a result, they are attributed to different school affiliations. Aren't these not recorded in the preface [to the *Sifen lü*]? What *Chao* 鈔 (i.e., Daoxuan's *Sifenlü shanfan buque xingshichao* 四分律刪繁補闕行事鈔) is based on is the Dharmaguptaka school. How could the person who has compiled [the vinaya of a Root] Section be the first patriarch of this [Dharmaguptaka] school? Thus the [Dharmaguptaka school we have] today should not base on this. This is the first reason for such an untenable [lineage assertion].

> 四師並以波離爲始祖，其所不可者三焉。且波離結集誦律，即今《僧祇》根本部也。《四分》一宗，雖從彼出，然派分已後，本枝競行，彼此相望，號爲異部，序不云乎？曇無德部，《鈔》者所宗，安有結集彼部之人，而預此宗之祖？此謂非今所宗，一不可也。[29]

According to Yuanzhao, the fundamental *vinayapiṭaka* compiled and recited by Upāli is the *Mohe sengqi lü* ascribed to a Root Section 僧祇根本部 in Indian Buddhism.[30] After the council of the five hundred saints, the denominational split in Indian Buddhism is rather complicated. Only the master who compiled the *Dharmaguptaka-vinaya* could be revered as the first patriarch of this school.

The second reason given by Yuanzhao is as follows: Moreover, though Upāli is credited with the achievements of compiling the vinaya, he is not the one transmitting it. In addition, Tanwude's master is Juduo 毱多 (i.e., Youbojuduo 優波毱多). [The learning of] Juduo could date back to Qieye 迦葉 (Skt. Kāśyapa). The genealogy [beginning with Kāśyapa] differs greatly from that [beginning with Upāli]. How could this be confused?

> 又，波離雖有結集之功, 不在傳法之數。況曇無德師本承毱多, 毱多已上, 至于迦葉, 師承頗異, 安可混同？[31]

According to the *Siji* 私記 (private record) of the *Mohe sengqi lü*,

> After the Buddha's *nirvāṇa*, Mahākāśyapa, who held eighty-four thousand *dharma* baskets compiled the *vinayapiṭaka* as to be the tenet of masters. After Mahākāśyapa's *nirvāṇa*, the elder Ānanda (Ch. Anan 阿難) also held eighty-four thousand *dharma* baskets, and then the elder Madhyāntika (Ch. Motiandi 末田地) also held eighty-four thousand *dharma* baskets, and then the elder Śāṇakavāsa (Ch. Shenaposi 舍那婆斯) also held eighty-four thousand *dharma* baskets. And then the

elder Upagupta, who was predicted by the Buddha to become the Buddha without the thirty-two or eighty marks (Skt. nirlakṣaṇa-buddha, alakṣaṇa-buddha; Ch. Wuxiang fo 無相佛), could not hold eighty-four thousand *dharma* baskets, as is said in the *Xiangmo yin yuan* 降魔因緣 (*Nidāna on Overcoming Demons*). Consequently, five divisions arose: the Dharmagupta (Ch. Tanmojueduo 曇摩崛多) being the earliest, then the Mahīśāsaka (Ch. Mishasai 彌沙塞) being the second, the Kāśyapīya (Ch. Jiayewei 迦葉維) being the third, the Sarvāstivāda (Ch. Sapoduo 薩婆多) being the fourth.

佛泥洹後，大迦葉集律藏爲大師宗，具持八萬法藏。大迦葉滅後，次尊者阿難亦具持八萬法藏，次尊者末田地，亦具持八萬法藏，次尊者舍那婆斯，亦具持八萬法藏，次尊者優波崛多，世尊記無相佛，如降魔因緣中説，而不能具持八萬法藏。於是遂有五部名生：初曇摩崛多別爲一部，次彌沙塞別爲一部，次迦葉維複爲一部，次薩婆多。[32]

In Sengyou's 僧佑 (445–518) *Chu sanzang ji ji* 出三藏記集 (*Collected Records concerning the Tripiṭaka*), similar stories of Mahākāśyapa and Upagupta are also told in its *Xinji lü fenwei wubu jilu* 新集律分爲五部記錄 (*Records on the newly compiled vinaya divided into five divisions*). Yuanzhao was influenced by these records and argued that Dharmagupta was not the successor to Upāli. His further analysis is as follows:

According to the *Datang nei dian lu* 大唐內典錄 (*A Catalog of The Buddhist Library in The Tang Dynasty*) [made] by Daoxuan, Upāli handed the *vinayapiṭaka* over to his disciple Dāsaka, Dāsaka handed it over to his disciple Sonaka, Sonaka handed it over to his disciple Siggava, Siggava handed it over to his disciple Tissa Moggaliputta. Tissa Moggaliputta handed it over to his disciple Zhantuobashe 旃陀跋闍 (Pā. Caṇḍavajji). The names of masters in the middle of this lineage are not evident. Finally, the *vinayapiṭaka* was handed over to Sengjiabaluo 僧伽跋羅 (Saṅghabhadra). It is known that Upāli started another lineage that merely promoted the *Sthavira-vinaya*. How could [Saṅghabhadra ascribed to] the Fazheng [bu] (i.e., the Dharmaguptaka school) get inserted into this lineage to inherit the *Sthavira-vinaya*?

又案南山《內典錄》云，波離以律藏付弟子陀寫俱，俱付須俱，須俱付悉伽婆，婆付目揵連子帝須，須付旃陀跋闍。中間不顯名氏，乃至付僧伽跋羅。是則波離別分一枝，專弘上座一律，安得橫以法正繼其後乎？[33]

This passage is mainly copied from the *Lidai sanbao ji* 歷代三寶記:

Upāli compiled the *vinayapiṭaka* after the Buddha's *nirvāṇa*. Immediately after that, on the fifteenth day of the seventh month of the same year, they held the *pravāraṇā* ceremony. They worshipped the *vinayapiṭaka* with fragrant flowers and made a dot at the front of the *vinayapiṭaka*. Year after year they did so. At the time Upāli was about to enter *nirvāṇa*, he handed the *vinayapiṭaka* over to his disciple Dāsaka. At the time Dāsaka was about to enter *nirvāṇa*, he handed the *vinayapiṭaka* over to his disciple Soṇaka. At the time Soṇaka was about to enter *nirvāṇa*, he handed the *vinayapiṭaka* over to his disciple Siggava. At the time Siggava was about to enter *nirvāṇa*, he handed the *vinayapiṭaka* over to his disciple Moggaliputta Tissa. At the time Moggaliputta Tissa was about to enter *nirvāṇa*, he handed the *vinayapiṭaka* over to his disciple Caṇḍavajji.[34] In this way, it was transmitted from master to master until the present *trepiṭaka* and *dharma* master. This *trepiṭaka* and *dharma* master arrived with the *vinayapiṭaka* in Guangzhou. Just before he was about to go on board a ship to return home and leave, he handed the *vinayapiṭaka* over to his disciple Saṅghabhadra. Saṅghabhadra, with the *śramaṇa* Sengyi translated the *Shanjian piposha* 善見毗婆沙 (i.e., the *Shanjianlü piposha*) in the sixth year of Yongming 永明六年 (488 A.D.) in the Zhulin Monastery 竹林寺in Guangzhou. On account of that, they stayed together for the rainy season retreat. Having held the *pravāraṇā* ceremony and worshipped the *vinayapiṭaka* with fragrant flowers at

midnight [on the 15th] of the seventh month, in the seventh year of Yongming 永明七年 (489 A.D.), they added a dot [to the record] as the former masters did.

佛涅槃後優波離既結集律藏訖，即於其年七月十五日受自恣竟，以香華供養律藏，便下一點置律藏前，年年如是。優波離欲涅槃，持付弟子陀寫俱; 陀寫俱欲涅槃, 付弟子須俱; 須俱欲涅槃, 付弟子悉伽婆; 悉伽婆欲涅槃, 付弟子目揵連子帝須; 目揵連子帝須欲涅槃, 付弟子旃陀跋闍。如是師師相付, 至今三藏法師。三藏法師將律藏至廣州臨上舶反還去, 以律藏付弟子僧伽跋陀羅。羅以永明六年（488）共沙門僧猗, 于廣州竹林寺譯出此《善見毗婆沙》。因共安居, 以永明七年（489）庚午歲七月半夜受自恣竟, 如前師法。以香華供養律藏訖，即下一點.[35]

"A dotted record of many sages 衆聖點記" is considered to be one of the most important historical sources for calculating the date of the historical Buddha. However, its authenticity was questioned as early as the Tang dynasty by Zhisheng 智昇, who emphasized that the *Shanjianlü piposha* is a vinaya commentary that interprets the tenets of a particular denomination and explains the outline 釋一家義，撮要而解 rather than an original vinaya canon recited by Upāli, and thus, it is not possible that the dotted record started from Upāli.[36] Yuanzhao also finds Fei Changfang's record questionable because Upāli and his later disciples merely promoted the *Sthavira-vinaya* 專弘上座一律, which means that Saṅghabhadra, a monk attributed to the Dharmaguptaka school, cannot be forcibly added to this genealogy. Influenced by his predecessors, Yuanzhao also identified the *Shanjianlü piposha* as a commentary on the *Sifen lü*. In his *Sifen lü xingshichao zi chi ji* 四分律行事鈔資持記 (*Commentary to Help Upholding the Vinaya for the Manual for Practice Based on the Sifen lü*), he says, "This vinaya commentary is composed by five hundred *arhat*s and is a commentary on the *Sifen lü*." 此論五百羅漢造, 釋《四分律》。[37] Here, "this vinaya commentary" refers to the *Shanjianlü piposha*. This narrative was copied from the *Sifen lü xingshichao jianzheng ji* 四分律行事鈔簡正記 (*A Collection of the Fine Comments from the Subcommentaries of the Sifen lü xing shi chao*) by Jingxiao 景霄, a monk from the Nanshan Vinaya school as well: The so-called *Shanjian* (i.e., the *Shanjianlü piposha*) means it is co-composed by five hundred *arhat*s and is a commentary on the *Sifen lü* 所言善見者, 謂五百羅漢共造, 斯論解《四分律》。[38] Among documents that predate Yuanzhao, this narrative is only seen in Jingxiao's work. They both claimed that the original Indic text of the *Shanjianlü piposha* arose during the First Council, but they gave no evidence to justify their claim. As far as I can see, the precise date of the *Samantapāsādikā* is also not mentioned in either the Pāli sources or Chinese translation. Nothing can demonstrate that any vinaya canon we see today came into being during the First Council. I assume Jingxiao's narrative is based on his own sectarian bias and reflects his emphasis on the orthodoxy in the Nanshan Vinaya school, which could be traced to the First Council. [39]

Yuanzhao further states,

In vinaya canons, it is Ānanda and Śāriputra (Ch. Shenzi 身子) that asked the Buddha about problems in the rules when the Buddha was alive. Besides, Śāriputra asked the Buddha to regulate rules, which was the beginning of the vinaya canons. His contribution is greater. Why isn't he the first patriarch? At the beginning of the *Sifen lü*, Upāli is called the beginner.[40] That is because the master of a section 部主 would compile the vinaya with a desire for all [five hundred] saints' verification. Upāli collected all helpful opinions from saints and thus is called a beginner. If one rigidly adheres to the literal meaning and makes Upāli the first patriarch, the five hundred saints [in the First Council] are all witnesses and participants to the compilation of the vinaya and thus should be the first patriarchs as well. Why is Upāli the only one to be the first? This is not transmission [of the vinaya]. This is the second reason for such an untenable [lineage assertion].

若謂佛世多所疑問者，律中阿難、身子請決尤多。況身子請佛制戒，爲發起之端，其功益大，何不爲祖？若謂律序初標波離爲首者，此乃部主將與集律，祈本

衆聖，以爲證信。而波離結集當衆之長，故言爲首耳。苟泥此文，必立爲祖，則餘身證者五百之衆，同是所祈，皆應爲祖，豈特波離乎？此謂不係傳襲。二不可也。[41]

The third reason given by Yuanzhao is as follows:

Moreover, if we refer to the *Fufazang yin yuan zhuan* with Indian origin, as well as Buddhist *sūtra*s and *abhidharma*s introduced into this kingdom, no schools would consider the one who compiled the canons as their first patriarch. If a certain first patriarch should be determined, it ought to follow the canonical corpora. For example, is Ānanda identified as the first patriarch [in any Buddhist text]? However, there is no such example.

又，歷觀西天《付法藏》，傳此土經論之家，未見取結集者爲祖。必如所立，亦應經宗，例以阿難爲祖邪？此無此例。[42]

The person who recited and compiled the texts in Buddhist councils is never treated as the first patriarch in Buddhist scriptures or historiographic works, such as the *Fufazang yinyuan zhuan*. As Yuanzhao said, if the one who recited and compiled the texts could be honored as the first patriarch, then Ānanda should be the choice, for the reason that Ānanda's listening to Buddha's teaching is a regular narration in Buddhist scriptures. There are many stories about Ānanda's direct learning from the Buddha about the regulation of rules in the vinaya canon as well. However, the schools of Huayan 華嚴, Tiantai 天台, Chan 禪, and other schools never regard Ānanda, who recited the texts, as their first patriarch. In the same case, Upāli cannot be revered as the first patriarch in the lineage of the Vinaya school. Therefore, Yuanzhao wrote the *Nanshan lüzong zucheng tulu* 南山律宗祖承圖錄 (*An Illustrated Catalogue of the lineage of the Nanshan Vinaya school*) to identify nine patriarchs of the Vinaya school, in which Dharmagupta is honored as the first patriarch. This was approved by contemporaries and, later, Buddhists. Gyōnen also quotes Yuanzhao's lineage of nine patriarchs in his narrative on the history of the Vinaya school in his *Risshū Kōyō* (Satō 1994, trans., p. 254).

## 4. Conclusions

This study pinpoints several aspects for further discussion.

Upāli is identified as the first patriarch in the patriarchal lineages of vinaya transmission in both the Pāli *Vinaya* commentary *Samantapāsādikā* and its parallel Chinese version, the *Shanjianlü piposha*, as well as in the *Mohe sengqi lü*, according to his reciting and compiling of the vinaya in the First Council. Yet, the latter two lineages were incorrectly interpreted by monks from the Vinaya school after the *Shanjianlü piposha* and *Mohe sengqi lü* were introduced into China and Japan.

The original Indian text of the *Shanjianlü piposha* was not known to Dingbin or other monks from the Vinaya school. Because the *Shanjianlü piposha* shares a very similar structure with that of the *Sifen lü* and both have corresponding passages, Dingbin misunderstood the *Shanjianlü piposha* as a commentary on the *Sifen lü*. In the patriarchal lineage recorded in the *Shanjianlü piposha*, there exists a "mistranslated" Tanwude, who is not found in its parallel in the Pāli sources. All these factors made Dingbin conceive this patriarchal lineage according to his sectarian bias.

Gyōnen followed Dingbin's assumption and further identified the patriarchal lineage in the *Shanjianlü piposha* as a patriarchal lineage of the Dharmaguptaka school. Apart from this, Fahu, in the patriarchal lineage in the *Mohe sengqi lü*, was also interpreted by Gyōnen as a patriarch transmitting the *Dharmaguptaka-vinaya*. That is to say, seen from Gyōnen's sectarian bias, any name that shares a literally synonymous meaning with Tanwude could be associated with the transmission of the *Dharmaguptaka-vinaya*. Both Dingbin and Gyōnen made great efforts to "present/promote" the Vinaya school with an orthodox Indian origin that could date back to Upāli. Because little learning about Indian Buddhism and original Indic texts was known to monks from the Vinaya school in medieval China and Japan,

Dingbin and Gyōnen could justify themselves in their narratives, and their explanations seemed convincing to those who knew little about Indian Buddhism.

During the Tang and Song dynasties, many Vinaya school monks studied both the Xiangbu and Nanshan Vinaya schools, although the former gradually merged into the latter in a later period. In the Northern Song dynasty, during the time when the Vinaya school had a temporary revival, Upāli was usually honored as the first patriarch in the various patriarchal genealogies of vinaya transmission constructed by eminent Chinese monks, which also revealed their will to orthodoxy. Yuanzhao, a renowned Nanshan Vinaya master of the same period, criticized this false construction, and he furthermore determined another patriarchal lineage of the transmission of the Vinaya school, in which the Indian patriarch Dharmagupta is made the first patriarch. This lineage determined by Yuanzhao also indicates his will to Indian Buddhist orthodoxy, and it receives the most attention, which both sectarian apologists and modern scholars have relied on.

Therefore, based on the narratives of monks from the Chinese and Japanese Vinaya schools, we can conclude that their own interpretations of the patriarchal lineages starting with Upāli in Indian vinaya texts that were later translated into Chinese are not historically reliable, while their orthodox construction of the patriarchal lineages beginning with Upāli, as well as later criticisms, fully display their limited knowledge of Indian Buddhism.

**Funding:** This research was funded by 國家社會科學基金重大項目《印度古典梵語文藝學重要文獻翻譯與研究》 (the major project of the National Social Science Foundation of China "Indian Art and Literary Theories in Classical Sanskrit Literatures: Translation and Studies on Fundamental Works (No. 18ZDA286)").

**Data Availability Statement:** All the data are calculated in this article, and there is no link.

**Acknowledgments:** I would like to thank Prof. Sasaki Shizuka for his helpful comments on an earlier draft of this article. I also express my sincere thanks to the anonymous reviewers for their valuable comments. But all errors and mistakes in the article are my own.

**Conflicts of Interest:** The author declares no conflict of interest.

## Abbreviations

Ch.　　Chinese
Jp.　　Japanese
Pā.　　Pāli
Skt.　　Sanskrit
*T*　　*Taishō shinshū daizōkyō*. Takakusu Junjirō 高楠順次郎 and Watanabe Kaigyoku 渡邊海旭 eds. *Taishō shinshū daizōkyō* 大正新修大蔵経 [*Buddhist Canon Compiled under the Taishō* Era (1912–1926)]. 100 vols. Tokyo: Taishō issaikyō kankōkai 大正一切経刊行会, 1924–1932.
*X*　　*Xinbian wanzi xu zangjing*. Nakano Tatsue 中野達慧, et al., eds. *Dai Nihon zokuzōkyō* 大日本續藏經 150 vols. Kyoto: Zokyō shoin, 1905–1912. Rpt. *Xinbian wanzi xu zangjing* 新編卍字續藏經 [*Buddhist Canon, Continued*] Taipei: Xinwenfeng, 1968–1978. Rpt, Chinese Buddhist Electronic Texts Association 中華電子佛典協會, CBETA Electronic Tripitaka Collection 電子佛典集成 Taipei: 1998–2018.

## Notes

[1]　Regarding the literature review on this issue, see Robson (2011).

[2]　Regarding this issue, see Wu (2018a, 2018b).

[3]　*Shanjianlü piposha* 2, *T* no. 1462: 24. 684b16-c11.

[4]　According to Jayawickrama, there is the lacuna of a *pāda* here. But the PTS version does not take this into account in the arrangement of the stanza. See Jayawickrama (1962, p. 181).

[5]　The Pāli passages and stanzas here are based on Takakusu and Nagai (1975, 2nd edition, pp. 61–63) and Jayawickrama (1962, pp. 181–82).

[6]　For an English translation, see Jayawickrama (1962, pp. 55–56). My translation is slightly different from Jayawickrama's.

7  Oldenberg (1982, 3rd edition, pp. 2–3). The same succession of vinaya masters is also recorded in the Pāli chronicles *Mahāvaṃsa* and the *Dīpavaṃsa*. For an investigation on this lineage's connections with inscriptions, Vincent Tournier shows how the epigraphic record of Āndhradeśa contains interesting clues with respect to the Tāmraparṇīya monks' self-representation, the echoes existing between inscriptions composed under their influence and the phraseology and terminology of Pāli Vinaya and historical writings. See Tournier (2018).

8  *Shanjianlü piposha* 5, *T* no. 1462: 24. 708a17–19. The parallel Pāli text reads, nānānāmā ti buddharakkhito dhammarakkhito tiādi nāmavasena vividhanāmā. nānāgottā ti gotamo moggallāno tiādi gottavasena vividhagottā. See Takakusu and Nagai (1975, 2nd edition, p. 187). Here, the Chinese text corresponds to the Pāli source.

9  *Shanjianlü piposha* 10, *T* no. 1462: 24. 740a18–23.

10  Regarding Dingbin's life biography, see Moro (2003) and L. Wang (2019).

11  According to the *Shanjianlü piposha*, there are twenty-four masters in the lineage from Tissa Moggaliputta and Mahinda to Sīva.

12  *Sifen lüshu shi zong yi ji* 3, *X* no. 733: 42. 41b1–21.

13  *Lidai sanbao ji* 5, *T* no. 2034: 49. 79b.

14  *Shanjianlü piposha* 2, *T* no. 1462: 24. 684b.

15  The *Aggikkhandhopama* has Chinese parallels in the *Mujiyu* 木積喻 (*T* no. 425) in the *Zhong ahan jing* 中阿含經, and in the *Kushu* 枯樹 (*T* no. 689) in the *Zengyi ahan jing* 增一阿含經.

16  The *Mahānāradakassapajātaka* has no Chinese parallels.

17  *Shanjianlü piposha* 16, *T* no. 1462: 24. 787a27–b12. My translation is slightly different from that of Bapat and Hirakawa.

18  In the English translation, Bapat and Hirakawa also give relevant numbers in this passage in the brackets (see Bapat and Hirakawa (1970, pp. 487–88)).

19  In the documents that predate the *Risshū Kōyō*, this saying only appears in Jingxiao's *Sifen lü xingshichao jian zheng ji*: Within the Sthaviravāda, there existed more sages and less ordinary persons. The Sthaviravāda remained in perfect harmony within two hundred years. At the beginning of the third century [after the Buddha's *nirvāṇa*], there was a little dissension and it was divided into two schools: 1. the Sarvāstivāda, 2. the [original] Sthaviravāda, which changed its name into the Haimavata school. Subsepuently 320 years [after the Buddha's *nirvāṇa*], one school named Vātsīputrīya issued from the Sarvāstivāda. Subsepuently 330 years [after the Buddha's *nirvāṇa*], four schools sprang from the Vātsīputrīya: 1. the Dhammottarīya, 2. the Bhadrāyaṇīya, 3. the Sammatīya, 4. the Channagirika. Subsepuently 360 years [after the Buddha's *nirvāṇa*], another school, the Mahīśāsaka, issued from the Sarvāstivāda. Subsepuently 380 years [after the Buddha's *nirvāṇa*], one school named the Dharmaguptaka (or called Fami bu) issued from the Mahīśāsaka. 其上座部，聖多凡小，二百年內，和合一味。至三百年初，有小乖諍，分爲二部，一説一切有部，二上座部轉名雪山部。次三百二十年，從一切有部，分出一部，名犢子部。次三百三十年後，從犢子部中，分出四部，一法上部，二賢胄部，三正量部，四蜜林山部。次三百六十年，從一切有部復分出一部，名化地部。次三百八十年，從化地部中，流出一部，名法藏部，或云法蜜。 See *Sifen lü xingshichao jian zheng ji* 1, *X* no. 737: 43. 21a10–b20. The cited passage deals with the divisions in the Sthaviravāda school, which Jingxiao mainly copies from the *Yibu zong lun lun* 異部宗輪論 (*A Treatise [called] the wheel of doctrines of different schools*) translated by Xuanzang 玄奘. However, Jingxiao's version is quite different from Xuanzang's translation in the dates of school divisions. According to Jingxiao, the Dharmaguptaka school emerged 380 years after the Buddha's *nirvāṇa*, while in Xuanzang's translation, it is stated, Immediately afterwards, during this third century, another school, the Mahīśāsaka, issued from the Sarvāstivāda. Immediately afterwards, during the same century, one school named the Dharmaguptaka issued from the Mahīśāsaka. 次後於此第三百年，從説一切有部，復出一部，名化地部。次後於此第三百年，從化地部流出一部，名法藏部。 See *T* no. 2031: 49. 15b14–16. For the English translation of this passage in the *Yibu zong lun lun* 異部宗輪論, see Masuda (1925, p. 16).

20  (Satō 1994, trans., pp. 234–35). This passage is also found in Gyōnen's *Risshū Gyōkanshō* 律宗瓊鑑章 (see 律宗瓊鑑章6, *dai nihon bukkyō zensho* 大日本仏教全書 105, p. 30).

21  *Sifen lü xingshichao sou xuan lu* 1, *X* no. 732: 41. 839b22–c2. Zhihong's *Sifen lü xingshichao sou xuan lu* is recorded in Eichō's 永超*Tōiki dentō mokuroku* 東域傳燈目録 (*Catalog of the Transmission of the Torch to the East*). That is to say, it was transmitted into Japan after the Tang dynasty. See *T* no. 2183: 55. 1156a2.

22  *Fufazang yin yuan zhuan* 3, *T* no. 2058: 50. 306a9–11. This passage about the division of five sects in the *Fufazang yin yuan zhuan* is extensively quoted in the donors' inscriptions in Dunhuang Cave 196. In addition, it is also stated in the donors' inscriptions in Dunhuang that the master of the *Sifen lü* is Tanwude. Regarding this issue, see Sheng (2017).

23  *Shanjianlü piposha* 2, *T* no. 1462: 24. 687a10–11. Here, it perfectly corresponds with its parallel in the Pāli sources. Regarding the dates of Mahinda and other vinaya masters in the lineage in the Pāli sources, see Mori (1984, pp. 455–56).

24  *Mohe sengqi lü* 32, *T* no. 1425: 22. 492c17–493a14.

25  The term *genben mohe sengqi* 根本摩訶僧祇 (the root text *Mohe sengqi*, or the *Mohe sengqi lü* ascribed to a Root Section) is also found in Yuanzhao's work. I will discuss it in the following note.

26  The term *genbenbu sifen lü* 根本部四分律 (the *Sifen lü* from a Root Section) reflects Yuanzhao's possible influence on Gyōnen. In his *Sifen lü xingshichao zi chi ji*, Yuanzhao states, From the Root Section, Venerable Fazheng edited and compiled the texts according to his own willing. Where he suspended his preach, there he marked with "one part 一分". [The texts from the Root

Section] was finally edited into a single volumn after he made such marks four times, thus this volunmn is called "Four-part vinaya". 以法正尊者於根本部中，隨己所樂，采集成文，隨說止處，即爲一分。凡經四番，一部方就，故號四分。 See *T* no. 1805: 40. 158a24–26.

27   Here, the authors of the *Zenghuiji* 增輝記主 possibly means the authors of the *Xingshichao zenghuiji* 行事鈔增暉記 (*A Zenghui Record on Daoxuan's Xingshichao*) (i.e., the vinaya master Huize 慧則 and his disiple Xijue 希覺 in Qianfo Monastery 千佛寺 in Qiantang 錢塘 in the period of Ten States 十國). See the *Xingshichao zhujiaji biaomu* 行事鈔諸家記標目 (*A Catalogue of Subcommentaries on Daoxuan's Xingshichao*), *X* no. 741: 44. 304c21–22. For an investigation on the *Zenghuiji*, see Zhan (2021).

28   *Zhiyuan yibian* 3, *X* no. 1104: 59. 647a5–12.

29   *Zhiyuan yibian* 3, *X* no. 1104: 59. 647a15–19.

30   The term *genben mohe sengqi* is also seen in the abovementioned Gyōnen work. Here, Yuanzhao's opinion can also be found in Daoxuan's *Sifenlü shanfan buque xingshichao* 四分律刪繁補闕行事鈔 (*the Sifen lü, Unnecessary Details Removed and Gaps Filled from Other Sources*): "The original texts [quoted here] means: The *Mohe sengqi lü* ascribed to a Root Section, and the others are ascribed to five divisions: 1. The Dharmaguptaka, that is the Four-Part Vinaya (*Sifen lü*), which the *Sifenlü shanfan buque xingshichao* is based on; 2. The Sarvāstivāda, that is the Vinaya of Ten Recitations (*Shisong lü*); 3. The Mahīśāsaka, that is the Five-part Vinaya (*Wufen lü*); 4. The Kāśyapīya, that is the Vinaya of Extrication (*Jietuo lü*, i.e., the *Jietuo jie jing* 解脫戒經), the *prātimokṣa* of which is existant; 5. The Vātsīputrīya whose vinaya has not come [to China]." 言正本者，《僧祇律》是根本部，餘是五部。曇無德部，《四分律》也，《鈔》者所宗。薩婆多部，《十誦律》也。彌沙塞部，《五分律》也。迦葉遺部，《解脫律》，此有戒本。婆麁富羅部，律本未至。See *Sifenlü shanfan buque xingshichao* 1, *T* no. 1804: 40. 3b23–25. Dajue 大覺 (dates of birth and death unknown), another monk from the Nanshan Vinaya school in the Tang dynasty, further argues in his *Sifenlü xingshichao pi* 四分律行事鈔批 (*A Critical Study on [Daoxuan's] Sifenlü Xingshi Chao*), "The *Mohe sengqi lü* is ascribed to a Root Section. The Sengqi school is called Mahāsaṃghika in the foreign language, here it is called 'Large community (dazhong 大眾)'. This means the council inside the city [of Rājagṛha], which is called the 'Section of the High-seated' with Kāśyapa as the leader. This is named after the senior age [of Kāśyapa]. Zhong 眾 means the group of five hundred saints, thus is called the 'Section of the Large Community'. This 'Section of the High-seated' is also called the 'Section of the Large Community', which is actually not the 'Section of the Great Community' gathering outside the city [of Rājagṛha]. The five divisions we have today derived from the former 'Section of the Large Community' organized by the High-seated, thus is called sengqi 僧祇. The 'Section of the Great Community' gathering outside the city is not the base of the *Sifen lü*. Therefore, the *Mohe sengqi lü* is identified as a root text of the five divisions. According to the *Dajijing* 大集經 (i.e., *Dafangdeng da ji jing* 大方等大集經), [the Buddha said, my disciples should] read extensively books of five divisions, which are thus called the Mohe sengqi. Here says "read extensively books of five divisions", that is to say, [the Mohe sengqi] is not any [certain division] of the five divisions, and so it is identified as a root section." 《僧祇律》是根本部者，僧祇部，外國云摩訶僧祇（Mahāsaṃghika），此云大眾。此是[ 王舍] 城內前結集者，名上座部，以迦葉在座年老得名也。眾既五百，名大眾部。呼此上座部爲大眾部耳，實非城外結集之大眾部也。今茲五部，皆從前上座之大眾部出，故呼僧祇。城外大眾部，非四分之根本也，所以將《僧祇》爲五部根本。據《大集經》云，廣博遍覽五部經書，是故名爲摩訶僧祇。既言遍覽五部，明知非五部數，故判爲根本部。See *Sifen lü xingshi chao pi* 1, *X* no. 736: 42. 623a9–16. According to Dajue, during the First Council, there existed two groups of saints: one group of five hundred saints with Mahākāśyapa as their leading elder who compiled the vinaya inside the city of Rājagṛha, in which Upāli recited it as the only systematic set of rules of the Buddha, and another group of one thousand saints who performed the compilation outside the city of Rājagṛha, which is called the "Section of the Great Community" due to the greater number of saints. The later five divisions are derived from the council in which the group of five hundred saints beginning with Mahākāśyapa gathered (i.e., the "Section of the Large Community organized by the High-seated 上座之大眾部" in Dajue's narrative). That is why the *Mohe sengqi lü* is regarded as a root text. This report, as far as I can see, is also repeated in Yuanzhao's *Sifen lü xingshichao zi chi ji* 四分律行事鈔資持記. (See *T* no. 1805: 40. 170a6–10.) But the expressions "compilation inside the city 城內結集" and "compilation outside the city 城外結集" only appear here in Dajue's work, while Yuanzhao states "compilation inside the [Pippala-]cave 窟內結集" and "compilation outside the [Pippala-]cave 窟外結集" instead. It seems that, here, the division between *Shangzuo* 上座 and *Dazhong* 大眾 was a natural one that occurred during the First Council rather than a schism, which only occurred around the events of the Second Council in Pāṭaliputra. Yuanzhao argues that what Upāli recited in the First Council is the root text the *Mohe sengqi lü*. It is quite possible that Dajue exerted an influence on Yuanzhao's identification. Therefore, regarding the terms *genben mohe sengqi* 根本摩訶僧祇 or *sengqi genbenbu* 僧祇根本部, it seems that Daoxuan, Dajue, Yuanzhao, and Gyōnen shared some common narrative lore, which indicates that they all assumed that the formation of the *Mohe sengqi lü* was earlier than vinaya texts attributed to other schools.

31   *Zhiyuan yibian* 3, *X* no. 1104: 59. 647a19–21.

32   *Mohe sengqi lü* 40, *T* no. 1425: 22. 548b9–17.

33   *Zhiyuan yibian* 3, *X* no. 1104: 59. 647a21–24.

34   Caṇḍavajji is treated as a disciple of Tissa Moggaliputta in the narrative of the *Lidai sanbao ji*. However, this Caṇḍavajji is Tissa Moggaliputta's teacher according to the *Shanjianlü piposha*: "Has learnt the line of succession of his teachers and has retained it without letting it slip from memory" means: Upāli learnt [the Vinaya] from the Tathāgata; Dāsaka learnt it from Upāli; Soṇaka from Dāsaka; Siggava from Soṇaka, Moggaliputta Tissa from Siggava and Caṇḍavajji. Thus the succession of teachers continues until it reaches the present. 次第從師受持不忘者，優波離從如來受，陀寫俱從優波離受，須提那俱從陀寫俱受，悉伽婆從須那

俱受，目揵連子帝須從悉伽婆受、又栴陀跋受，如是師師相承，乃至於今。See *Shanjianlü piposha* 6, *T* no. 1462: 24. 716c25–29. Here, both Zhantuoba 栴陀跋 and Zhantuobashe 旃陀跋闍 are transliterations of Caṇḍavajji. The *Shanjianlü piposha* relates the same as the Pāli *Samantapāsādikā* and chronicles: namely, that Caṇḍavajji was the teacher of Moggaliputta Tissa, not his successor. W. Pachow has pointed out that the sixth name Caṇḍavajji that Fei Changfang gave here is a mistake. See Pachow (1965).

35　*Lidai sanbao ji* 11, *T* no. 2034: 49. 95b20–c6.

36　*Kaiyuan shijiao lu* 開元釋教錄 (*Record of Śākyamuni's Teachings Compiled During the Kaiyuan period*) 6, *T* no. 2154: 55. 536a7–9.

37　*Sifen lü xingshichao zi chi ji* 4, *T* no. 1805: 40. 170b4. It is interesting to note Yuanzhao's contradiction in interpreting the vinaya canon compiled/composed in the First Council. Here, he claimed that the original Indic text of *Shanjianlü piposha* was composed by five hundred *arhat*s in the First Council. However, according to the *Zhiyuan yibian*, as shown in the abovementioned passages, he stated that the fundamental *vinayapiṭaka* compiled and recited by Upāli is the present *Mohe sengqi lü* ascribed to a Root Section, which the later *Dharmaguptaka-vinaya* was derived from. The contradiction here is obvious: because Yuanzhao classified the *Shanjianlü piposha* as a *Dharmaguptaka-vinaya* commentary made in the First Council, how could the date of a vinaya commentary be much earlier than the vinaya texts it comments on?

38　*Sifen lü xingshichao jianzheng ji* 4, *X* no. 737: 43. 57b10–11.

39　For a full discussion on this, see (Wu 2018a).

40　In the verses at the beginning of the *Sifen lü*, it is said: Upāli is the beginner, with other witnesses and participants [in the First Council]. Now the outline of rules should be told, listened by all saints. 優波離爲首，及餘身證者；今說戒要義，諸賢咸共聽。See *Sifen lü* 1,*T* no.1428: 22. 567b28–c1.

41　*Zhiyuan yibian* 3, *X* no. 1104: 59. 647b1–7.

42　*Zhiyuan yibian* 3, *X* no. 1104: 59. 647b7–10.

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
