# Peer review of "On the Patriarchal Lineages of Vinaya Transmission Starting with Upāli: Narratives and Interpretations in the Vinaya School 律宗 in China and Japan"

_religions, doi:10.3390/rel14040464_

Round 1

Reviewer 1 Report

I think the author has done a very thorough examination of the primary and secondary resources. The textual analysis is very detailed and commendable. My major issue is that I do not see a clear research question and the academic contribution of this article. I do not mean that the author has not clarified "Vinaya school masters' sectarian views of Indian Buddhism" and "a deeper understanding on the development of the Vinaya school in China and Japan." What I could not figure out is why it is important for us to know about such sectarian views and the development of Vinaya school. Is it important because it tells us how clerics creatively interpret doctrinal resources to show their agency in the transcultural context? Is it important because it tells us how creative interpretation of vinaya facilitates these clerics' effort in community building?  I hope the author can make it easier for the readers to see the point more directly.

Reviewer 2 Report

Overall this is a very interesting and informative article. Thank you! My main recommendation for improving the article is that you rewrite the introduction. Rather than starting off by discussing Upāli, it would be better I think to highlight your main contribution, namely how Chinese and Japanese Buddhist scholars constructed Vinaya lineages with very little understanding of Indian Buddhism. Having set out the article's main thesis, then you can segue to discussing Upāli. Also, the article variously uses Sanskrit and Pali terms. When discussing place names such as Jambudvīpa/Jambudīpa, it might be a better idea to use the Sanskrit terms since these seem to be better known. Some other minor issues include the following:

Line 38

Change “hardly” to “very little”

Line 47

Change “interpret” to “interpreted”

Line 84

Change “held” to “held the”

Line 136

Correct formatting in this line.

Line 138

Change “It is based on the previous research that,” to “It is on the basis of the previous research that”

Lines 145, 149 et al.

By “Jambudīpa” do you mean “Jambudvīpa,” the central human contintent?

Lines 153, 157 et al.

By “Sīhaḷadīpa” do you mean “Siṅhaladvīpa,” the island of Ceylon?

Line 180

Change “Sīva” to “Śiva”?

Lines 181-233

I’d recommend that the long primary source texts be moved to an appendix.

Line 237

 “Jambusiri” should be read as “Jambuśrī,” an alternate name for “Jambudvīpa”?

Line 278

Omit word “either”

Line 282

Change “to” to “about”

Line 732

Change “totally recorded” to “evident”

Line 847

Change “distortedly” to incorrectly”?

Reviewer 3 Report

This is an important and well argued study on vinaya lineage formation which shows that the establishment of Tanwude as a patriarch was likely a creation of Chinese vinaya monks who attempted to establish the allegiance of the Dhramaguptaka vinaya school to an authentic Indian root.  

It could be interesting, here or on other occasion, to consider the vinaya lineage in the donor’s inscription contained in the Dunhuang cave 196. This 9thcentury inscription claims  lineage from the Buddha to his immediate disciples, and their followers are distributed into five categories in accordance with their different capacities  of upholding the vinaya. According to the inscription, the vinaya transmission to China began with the imperial translation project led by the ruler Yao Xing (366-416) in  410 in  Chang’an. This could be interesting to compare these claims with standard literary narratives that are discussed in the paper.

I have noticed a few minor issues related to some conventions. For instance, Songshan Mountain in line 379 should be either Mount Song or just Songshan to correctly correlate to its name in Chinese. Also, abbreviation for “Chinese” in brackets (for instance, line 145) is referred to as (Chin.). It is usually accepted to use (Chinese/Ch.:…) on the first occasion, and then just (Ch.:). 

Publication recommended with minor revisions.
